# Epidermal growth factor-like domain 7 drives brain lymphatic endothelial cell development through integrin αvβ3

Jingying Chen [1] ✉, Jing Ding[2], Yongyu Li[2], Fujuan Feng[2], Yuhang Xu[2], Tao Wang[2], Jianbo He[2], Jing Cang[1] & Lingfei Luo [1,2] ✉

In zebrafish, brain lymphatic endothelial cells (BLECs) are essential for meningeal angiogenesis and cerebrovascular regeneration. Although epidermal growth factor-like domain 7 (Egfl7) has been reported to act as a pro-angiogenic factor, its roles in lymphangiogenesis remain unclear. Here, we show that Egfl7 is expressed in both blood and lymphatic endothelial cells. We generate an *egfl7 cq180* mutant with a 13-bp-deletion in exon 3 leading to reduced expression of Egfl7. The *egfl7 cq180* mutant zebrafish exhibit defective formation of BLEC bilateral loop-like structures, although trunk and facial lymphatic development remains unaffected. Moreover, while the *egfl7 cq180* mutant displays normal BLEC lineage specification, the migration and proliferation of these cells are impaired. Additionally, we identify integrin αvβ3 as the receptor for Egfl7. αvβ3 is expressed in the CVP and sprouting BLECs, and blocking this integrin inhibits the formation of BLEC bilateral loop-like structures. Thus, this study identifies a role for Egfl7 in BLEC development that is mediated through the integrin αvβ3.

The lymphatic vessels, including the meningeal lymphatic vessels, play physiological roles in fluid homeostasis, fat and macromolecule absorption, drainage of waste, and immune surveillance[1]. In vertebrates, the lymphatic endothelial cell (LEC) specification and maintenance depend on Prox1[2,3], and embryonic lymphangiogenesis is driven by VEGFC signaling via VEGFR3[4]. The matrix protein Ccbe1 acts with metalloprotease Adamts3 to generate mature and biologically active VEGF-C protein[5–7]. Integrin β1 is another receptor expressed by LECs that contributes to lymphangiogenesis, and it can modulate the intracellular phosphorylation and activation of VEGFR3 kinase in response to the mechanical stimulus[8,9].

Zebrafish is an excellent model for high-resolution, in vivo live imaging of lymphangiogenesis[10,11]. The trunk lymphangioblasts first sprout from posterior cardinal veins (PCV) and migrate to the horizontal myoseptum (HM). These parachordal lymphangioblasts (PL) then migrate ventrally and dorsally to remodel into the thoracic duct (TD) and the dorsal longitudinal lymphatic vessel (DLLV)[12]. The facial

lymphatic sprout (FLS) differs from the previously characterized trunk lymphatics. It arises from the common cardinal vein (CCV) and other non-venous progenitors[13], then migrates to form the lateral facial lymphatic vessel (LFL), the otolithic lymphatic vessel (OLV), the medial facial lymphatic (MFL), and the lymphatic branchial arches (LAAs)[14].

Zebrafish also harbour brain lymphatic endothelial cells (BLECs) also known as fluorescent granular perithelial cells (FGPs) or meningeal mural lymphatic endothelial cells (muLECs). These cells show the same gene expression signature as other LECs but do not form vessels. Instead, they form a network of individual cells covering the brain surface[15–17]. This is different from meningeal lymphatic vessels, which attach to the inner surface of the skull[18]. These BLECs are derived from the choroidal vascular plexus (CVP) at 56 h post-fertilization (hpf) and migrate along the mesencephalic vein (MsV) to form a bilateral loop over the optic tectum at 5 days post-fertilization (dpf). Moreover, the BLECs are crucial in regulating meningeal angiogenesis and managing severe pathological processes like ischemic stroke. After brain vascular

[1]School of Life Sciences, Department of Anaesthesia of Zhongshan Hospital, Fudan University, 200438 Shanghai, China. [2]Institute of Developmental Biology and Regenerative Medicine, Southwest University, 400715 Chongqing, China. ✉e-mail: jingyingchen@fudan.edu.cn; lluo@fudan.edu.cn

injury, BLECs formed lumenized lymphatic vessels and grew into the parenchyma, directed by Cxcl12b/Cxcr4a[19]. These ingrown lymphatic vessels (iLVs), on the one hand, drain interstitial fluid to resolve edema; on the other hand, they transdifferentiate into early-regenerated blood vessels and act as a "growing track" for nascent blood vessels[20,21]. Consistent with their lymphatic nature, the development of BLECs depends on the Vegfc/Vegfd/Vegfr3/Ccbe1 signaling pathway. However, whether some new molecular mechanisms regulate BLECs forming a vessel-like loop during the initial sprouting is still unknown.

EGFL7 is a highly conserved secreted angiogenic factor found in vertebrates. This extracellular matrix-bound factor is expressed almost exclusively by endothelial cells and has a specific role in blood vessel development by influencing the extra cellular matrix (ECM) environment[22]. EGFL7 consists of various putative protein domains, including an elastin microfibril interface (EMI) part and two centrally located EGF-like domains[23]. It is a specific ligand for integrin on endothelial cells and is strongly associated with fibronectin in the ECM[24]. Several studies have shown that knockdown of *egfl7* in zebrafish, frogs, and HUVECs suppresses endothelial cell proliferation, adhesion, migration, and vascular tube formation[25–27]. However, the *egfl7* mutant of zebrafish and one Egfl7-specific knockout mouse line lack obvious phenotypes, which can be explained by activation of compensatory genes or the biological effects of miR-126[28,29]. EGFL7 also plays a role in vascular repair, CNS inflammation, tumor metastasis, and cancer angiogenesis[30–33]. However, it remains unclear whether it plays a role in lymphangiogenesis despite its critical role in angiogenesis.

In this study, we aim to determine whether Egfl7 is also important for BLEC development in zebrafish. We find that Egfl7 plays a role in the early stages of lymphangiogenesis by forming a vessel-like loop via migration and proliferation in the brain, either autonomously or non-autonomously. However, it is not necessary for the specification of BLECs. Although BLECs sprouting depends on Vegfc, Vegfd, and Ccbe1 signaling, Vegfc overexpression hardly rescues the devoid of BLECs caused by *egfl7* mutation. Integrin αvβ3 is a specific receptor of EGFL7 found in the CVP and departing BLECs. We observed that the inhibition of integrin αvβ3 impaired BLECs migration. Furthermore, the suppression of integrin-linked kinase (ILK), which regulates VEGFR3 signaling by controlling its interaction with integrin, partially rescues the complete loss of lymphatic loops observed following *egfl7* depletion. Our findings suggest that Egfl7 plays a role in regulating brain lymphatic development through integrin αvβ3.

## Results

### Egfl7 is required for BLEC formation and brain vascular regeneration

Given that Egfl7 is a critical secreted pro-angiogenic factor and is nearly restricted to the endothelial cells during embryogenesis as well as physiologic and pathologic angiogenesis, we examined whether Egfl7 plays roles in brain vascular regeneration. In the *Tg(kdrl: DenNTR)* transgenic nitroreductase (NTR)-metronidazole (Mtz) cerebrovascular injury model, BLECs were activated in response to the injuries and quickly ingrew into the injured parenchyma to form lumenized LVs[20]. Using whole-mount in situ hybridization and fluorescent in situ hybridization (FISH) combined with antibody staining, we found expression of *egfl7* in all the endothelium, including BLECs, iLVs, and blood vascular endothelial cells (BECs), regardless of injuries or not (Supplementary Fig. 1). Furthermore, the *Tg(egfl7:YFP)^cq181* transgenic line was generated to confirm the expression of *egfl7* in the endothelium of brain and trunk (Fig. 1). These data suggest the expression of *egfl7* in both lymphatic and blood vessels.

We next examined lymphatic development and brain vascular regeneration in the *egfl7* mutant. The *egfl7^cq180* mutant was generated by CRISPR/Cas9, with a 13-bp-deletion in the exon 3, leading to truncation of the EMI domain and a frameshift from amino acid 51 thereby a

premature stop codon at amino acid 71 (Supplementary Fig. 2a). This mutation caused nonsense decay of *egfl7* mRNA and absence of protein expression (Supplementary Fig. 2b, c). It was similar to the *egfl7^s981*, a well-established mutant allele. Unlike *egfl7* morphants that exhibit severe vascular tube formation, *egfl7^s981* displays normal gross morphologies in vasculogenesis and angiogenesis[29]. This *egfl7^cq180* mutant exhibited no obvious abnormalities in blood vessel development, but the *lyve1b* + BLECs failed to populate into the bilateral loop-like structures from 4dpf to 6dpf (Fig. 2a–c). To confirm the defective BLEC development, the expression of lymphatic markers including *vegfr3*, *mrc1a*, and *prox1*, was analyzed. Downregulations of *vegfr3* and *mrc1a* were detected in the brain of the mutant (Fig. 2f–j). And the *prox1a* promoter-driven transgenes[34] validated the lack of BLEC bilateral loop-like structures in the mutant (Supplementary Fig. 3a–d). In addition, we examine whether the devoid of BLEC in the mutant could cause defects in brain vascular regeneration. After cerebrovascular injury, the iLV ingrowth and nascent BV regeneration did not occur in the *egfl7^cq180* (Supplementary Fig. 3e, f). Consequently, the majority of injured mutants experienced brain edema and died within ten days after Mtz treatment (Supplementary Fig. 3g, h).

Although the *egfl7^cq180* displayed delayed, transient, mild reduction of lymphatic sprouts in OLV, MFL, LFL, and PLs before 4 dpf (Supplementary Fig. 4a–h), the facial lymphatics, TD, and DLLV became recovered by 6 dpf (Fig. 2a, b, d, e). Furthermore, the *egfl7^cq180* adults failed to form regular meningeal blood vessel network, which might be caused by the near absence of the *lyve1b* + BLECs (Fig. 2k–p). Despite that, the *egfl7^cq180* adults exhibited overall normal morphologies, and became viable and fertile (Supplementary Fig. 4i). These data indicate that *egfl7* is required for the BLEC development, but not for the development of facial and trunk lymphatics. Furthermore, BLECs are not necessary for the viability and fertility of zebrafish.

The *mir-126b* is located in the intron 6 of *egfl7*, and has been implicated in lymphatic development and vascular integrity[28,35,36]. To investigate whether the lack of BLEC and defects in blood vascular regeneration were attributed to the loss of *mir-126* function, we analyzed the expression levels of *mir-126*. Real-time RT-PCRs of WT and *egfl7^cq180* demonstrate comparable *mir-126* expression levels (Supplementary Fig. 4j), indicating that the BLEC phenotypes in the *egfl7* mutant are independent of *mir-126*.

Egfl7 expressed in both LECs and BECs. To test the roles of Egfl7 in BLEC development is cell autonomous or not, we generated two stable transgenic lines *Tg(kdrl:egfl7-p2A-GFP)^cq182* and *Tg(lyve1b:egfl7-p2A-GFP)^cq183* under the mutant background to specifically replenish Egfl7 in the BECs and LECs, respectively. In contrast to the *Tg(kdrl: GFP)* or *Tg(lyve1b: GFP)* control group, replenishing Egfl7 in either BECs or LECs could rescue the formation of BLEC bilateral loop-like structures in *egfl7^cq180* mutants at 6 dpf (Fig. 3b, d, e, g, i, j). These results indicate that both the BEC-derived and LEC-derived Egfl7 are functional for BLEC formation. Additionally, overexpression of Egfl7 in the wild-type BECs using the *Tg(kdrl:egfl7-p2A-GFP)* transgene induced ectopic BLEC formation in contrast to overexpression of GFP using the *Tg(kdrl: GFP)* transgene (Fig. 3a, c, e). Whereas Egfl7 overexpression in the wile-type LECs is ineffective to the number of BLECs (Fig. 3f, h, j). This should be caused by more abundant meningeal blood vessels than BLECs in the top layer, so more BLECs were induced by the blood vessel-derived Egfl7. All these data suggest that the endothelial cells-derived Egfl7 is required for the BLEC development.

### Egfl7 is dispensable for specification but necessary for migration and proliferation of BLECs

From about 56 hpf, the *vegfr3*-positive and low-level *kdrl*-expressing BLEC progenitors sprout from the CVP, then proliferate and migrate along the MsV to form bilateral loop-like structures of mural lymphatic cells over the brain surface at 3 dpf[37]. The BLEC progenitors also express the lymphatic marker *prox1*[15]. To investigate whether the BLEC

progenitors could be specified from the CVP in the *egfl7 cq180*, we examined Prox1, *vegfr3*, and *mrc1a* expression as the markers of LEC fate. We performed anti-Prox1 antibody staining in the *lyve1b* reporter line, in which the Prox1-positive nuclei were detectable in the lymphatic sprouts from CVP in the *egfl7 cq180* at 3 dpf (Fig. 4a–d). Furthermore, combination of FISH and antibody staining indicated that the *lyve1b⁺kdrl*low BLEC progenitors in the mutant co-expressed *vegfr3* and *mrc1a* (Fig. 4e–i, arrowheads). However, although the induction of lymphatic identity maintained at the CVP of *egfl7 cq180*, the number of *vegfr3+* and *mrc1a+* BLECs significantly reduced (Fig. 4j).

Because BLEC specification was unaffected in the mutant, we next investigate whether the defective BLEC development was caused by defects in cell migration or proliferation. The time-lapse imaging was performed to illustrate the process of loop-like structure formation using the *Tg(lyve1b: DsRed; kdrl: DenNTR)* transgenic line. In the wildtype, the BLEC progenitors sprouted from the CVP in the lower middle layer of the brain, then migrated along the MsV, which is located in the

top layer of the brain, to form the loop-like structures (Fig. 5a, b and Supplementary Movie. 1). By contrast, in the *egfl7* mutant, most of the sprouting cells stopped around CVP and failed to migrate, and formation of bilateral loop-like structures that covered the optic tectum was blocked (Fig. 5a, c and Supplementary Movie. 1). Furthermore, we analyze the expression of proliferating cell nuclear antigen (PCNA) in BLECs that shows individual cells outside of the G0-phase of the cell cycle. At 3 dpf, the *lyve1b* + BLECs exhibited high proliferation in the control larvae, but rarely proliferated in the mutant (Supplementary Fig. 5a–d). Injection of EdU (5-ethynyl-2′-deoxyuridine) confirmed that BLEC proliferation became significantly reduced in the mutant (Fig. 5d–f and Supplementary Fig. 5e–j). The terminal deoxynucleotidyl transferase-mediated deoxyuridine triphosphate nick-end labeling (TUNEL) assays showed that TUNEL and *lyve1b* double-positive cells was hardly detectable in the *egfl7* mutant (Supplementary Fig. 6). All these data demonstrate that the *egfl7* mutation blocks proliferation and migration of BLECs, but does not induce their apoptosis.

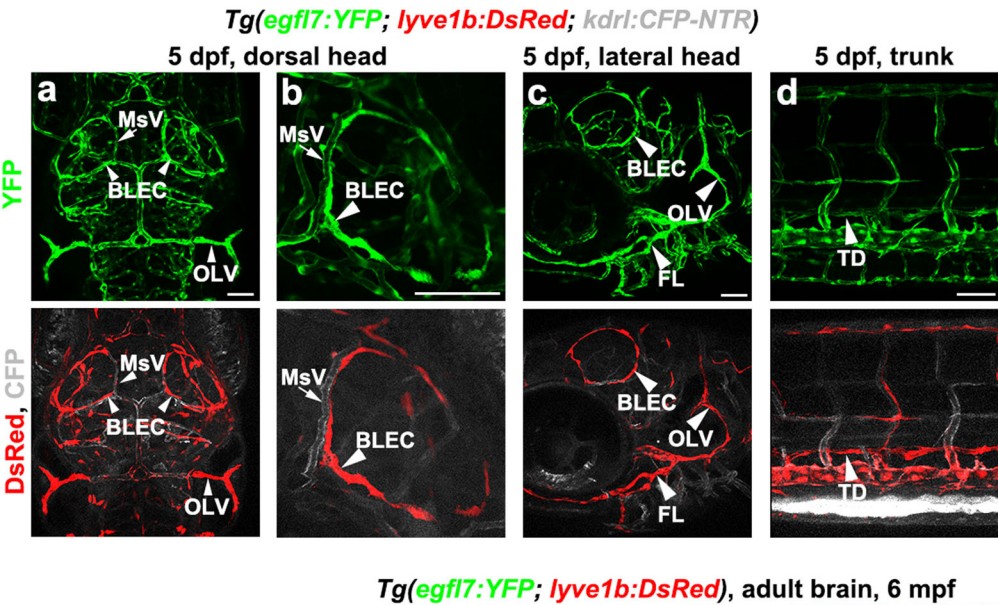

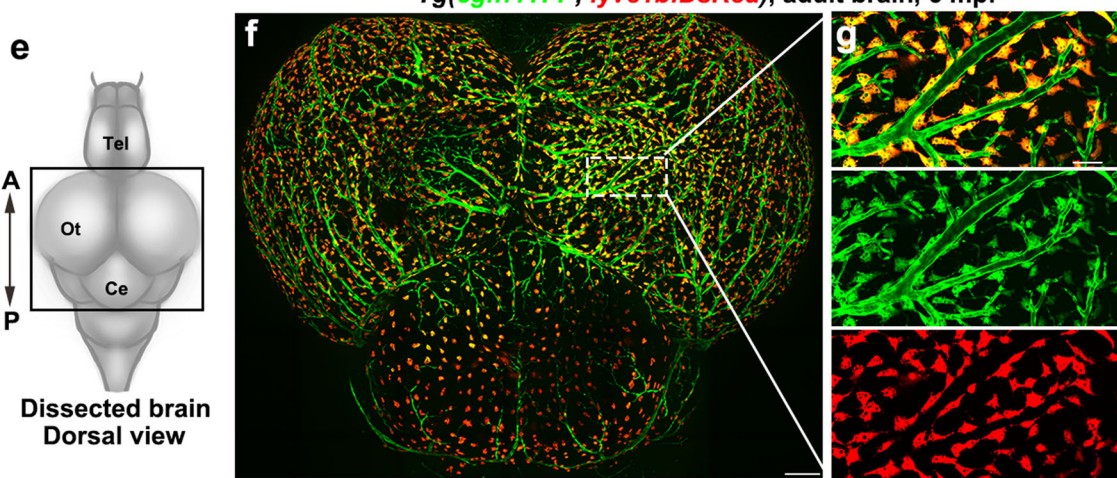

**Fig. 1 | *Tg(egfl7:YFP)* transgenic zebrafish express YFP in blood vessels and lymphatics. a–d** *Tg(egfl7:YFP)* was generated showing YFP in blood vessels and lymphatics. The distribution of YFP is similar with CFP and DsRed in the triple transgenic line *Tg(egfl7:YFP; lyve1b: DsRed; kdrl: CFP-NTR)* at 5 dpf. *n* = 25/25 embryos. The low-magnified image shows the dorsal view of the head (**a**), the high-magnified image in another embryo shows the magnified BLECs loop in the dorsal brain (**b**). The lateral facial lymphatics of the head shows in **c**. The trunk lymphatics shows in **d**. BLECs, brain lymphatic endothelial cells, MsV, mesencephalic vein, OLV, otolithic lymphatic vessel, FL, facial lymphatics, TD, thoracic duct. Scale bar, 50 μm. **e** Schematic diagram showing the dorsal view of a dissected adult zebrafish brain. **f** *egfl7* promoter-driven YFP is expressed in all BLECs and blood vessels. *n* = 6/8 adults. **g** High-magnification inset showing the YFP overlap with DsRed in the double transgenic line *Tg(egfl7:YFP; lyve1b: DsRed)*. Scale bars, 200 μm in **f** and 50 μm in **g**.

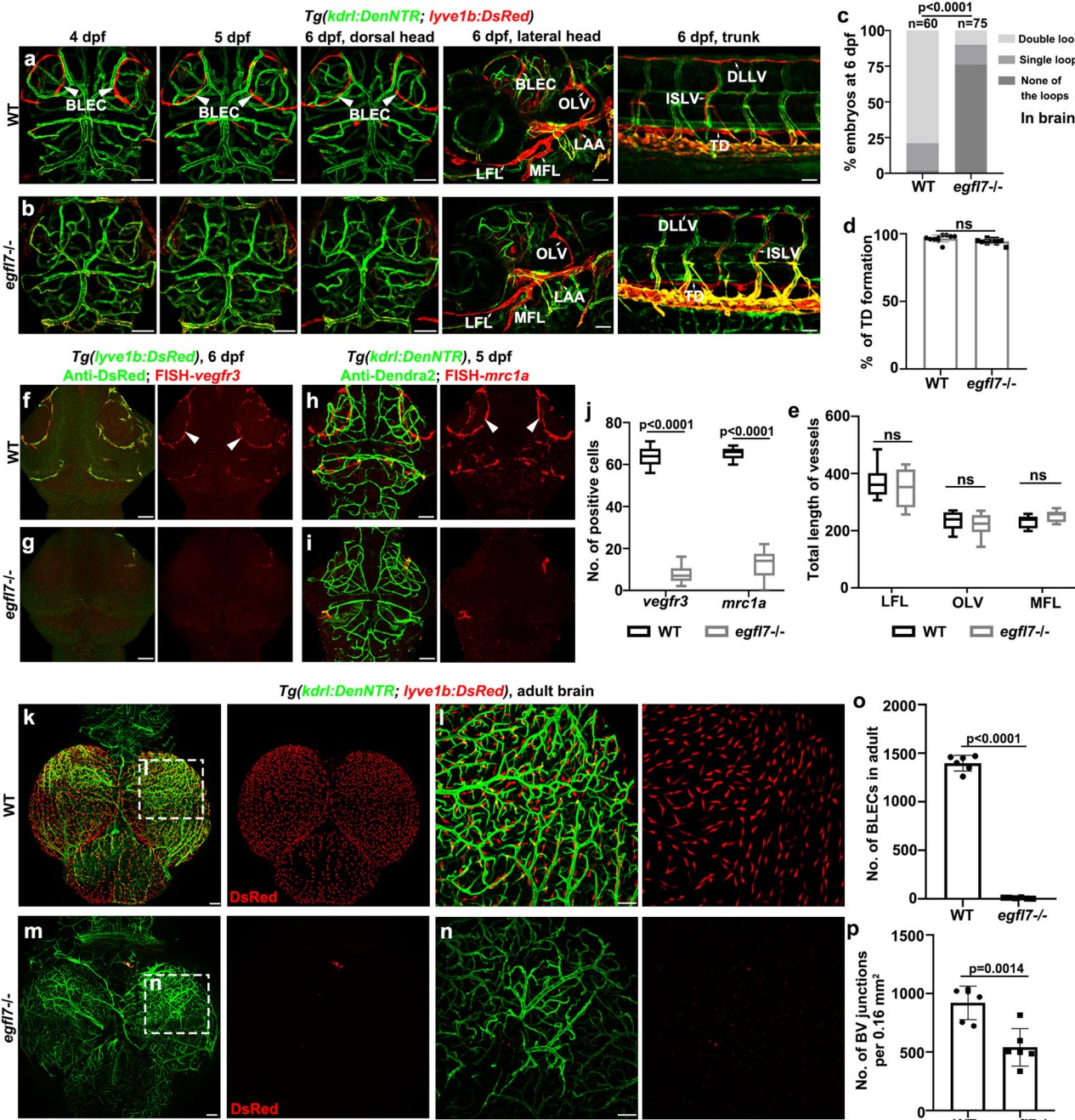

**Fig. 2 | *egfl7* is essential for zebrafish BLEC formation. a** Confocal images of the BLECs, facial lymphatics, and trunk lymphatics in *Tg(kdrl: DenNTR;lyve1b: DsRed)* transgenic lines from 4 dpf to 6 dpf in WT. **b** Loss of *egfl7* prevents the formation of the BLECs from 4 dpf to 6 dpf, but the facial lymphatics of the lateral head and trunk lymphatics are normal in *egfl7* mutant embryos at 6 dpf. **c** Percentage of embryos that have double lymphatic loops, single loops, and none of the loops in the brain (WT, *n* = 60 embryos, *egfl7*-/-, *n* = 75 embryos, χ² test). **d** The statistics show the percentage of TD formation per 6 somites (*n* = 10 embryos, two-tailed unpaired *t*-test; ns, no significance. Data are represented as mean ± SD). **e** The statistics show the total lengths of LFLs, OLVs, and MFLs at 6 dpf (*n* = 8 embryos, 2way ANOVA multiple comparisons test; ns, no significance. Box plots show the five-number summary of a set of data: including the minimum score (shown at the end of the lower whisker), first (lower) quartile, median, third (upper) quartile, and maximum score (shown at the end of the upper whisker)). BLECs, brain lymphatic endothelial cells; LFL lateral facial lymphatic, MFL medial facial lymphatic, LAA lymphatic branchial arches, OLV otolithic lymphatic vessel, ISLV intersegmental lymphatic

vessels, DLLV dorsal longitudinal lymphatic vessel, TD thoracic duct. **f**, **g** Double labeling of FISH-*vegfr3* and anti-DsRed in *Tg(lyve1b:DsRed)* transgenic background at 6 dpf. Arrowheads point to the BLECs in the top layer of the brain. **h**, **i** Double labeling of FISH-*mrc1a* and anti-Dendra2 in *Tg(kdrl:DenNTR)* transgenic background at 5 dpf. **j** The statistics show the number of *vegfr3*+ and *mrc1a* + BLECs in WT and *egfl7* mutant (*n* = 9 embryos, 2-way ANOVA multiple comparisons test. Box plots show the five-number summary of a set of data: including the minimum score (shown at the end of the lower whisker), first (lower) quartile, median, third (upper) quartile, and maximum score (shown at the end of the upper whisker). **k–n** Dissection and whole-mount images of BLECs and meningeal vascular in *Tg(kdrl: DenNTR;lyve1b: DsRed)* adult brain at 6 mpf (months post-fertilization) in WT (**k**, **l**) and *egfl7* mutant (**m**, **n**). **o**, **p** The statistics show the number of *lyve1b* + BLECs in the adult brain (**o**, *n* = 6 brains; two-tailed unpaired *t* test) and the number of BV junctions per 0.16 mm² of a brain (**p**, *n* = 6 brain areas; two-tailed unpaired *t* test). Data are represented as mean ± SD. Scale bar, 50 μm.

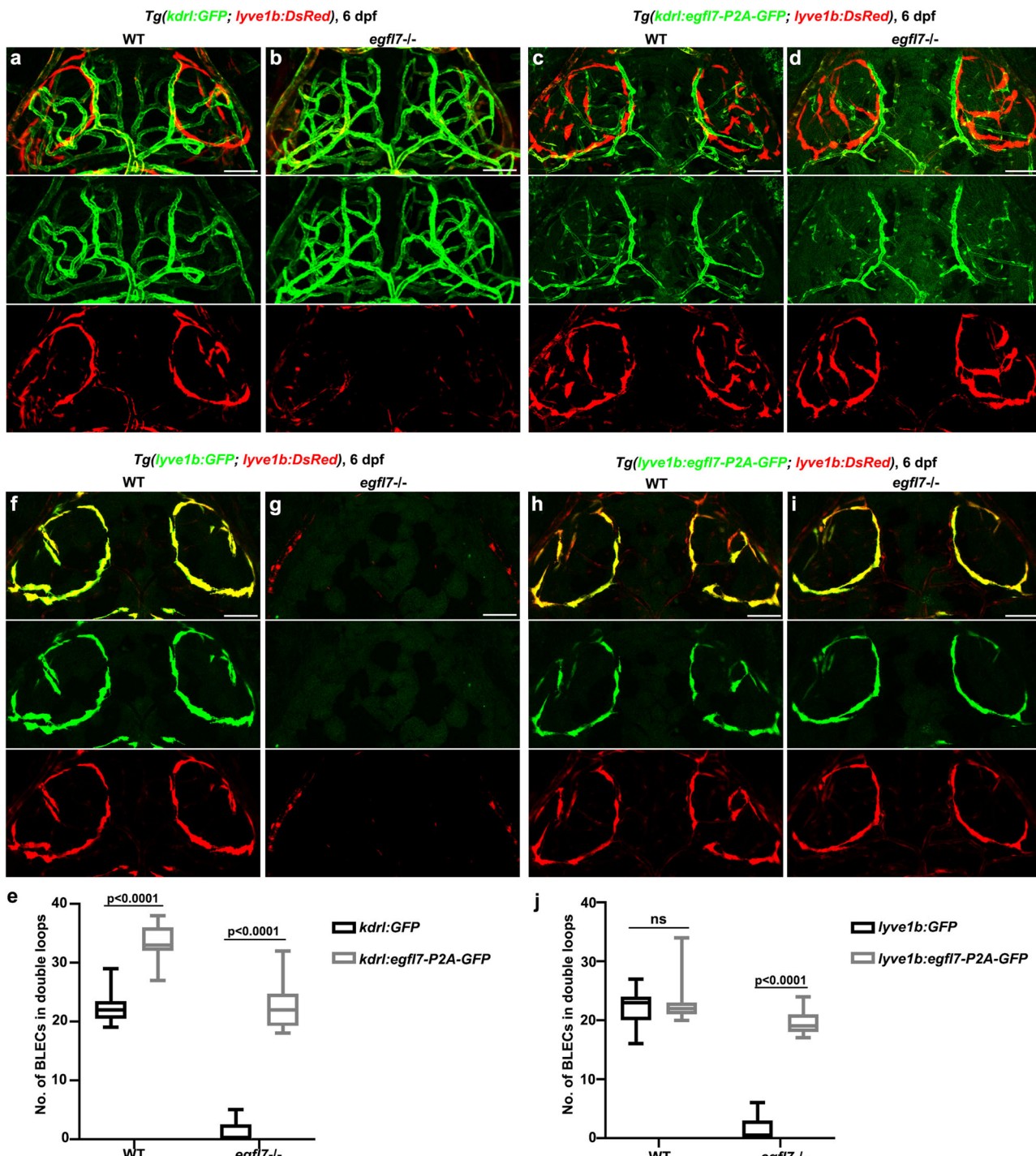

**Fig. 3 | *egfl7* regulates brain lymphatics development either autonomously or non-autonomously. a–d** In the stable transgenic line *Tg(kdrl:egfl7-p2A-GFP)*, the replenishment of Egfl7 in BVs can rescue the absence of BLECs in the *egfl7* mutant (**d**) in contrast to the mutant under the *Tg(kdrl: GFP)* transgenic line (**b**). For WT, overexpression of Egfl7 in the BVs results in increased BLECs emerge in the bilateral loop over the brain (**c**) compared to the WT under the *Tg(kdrl:GFP)* (**a**). **e** The statistics show the number of BLECs per double loops in WT and *egfl7* mutant under *Tg(kdrl: GFP)* and *Tg(kdrl:egfl7-p2A-GFP)* transgenic lines (*n* = 24 embryos, 2-way ANOVA. Box plots show the five-number summary of a set of data: including the minimum score (shown at the end of the lower whisker), first (lower) quartile,

median, third (upper) quartile, and maximum score (shown at the end of the upper whisker)). **f–i** In contrast to the mutant under *Tg(lyve1b: GFP)* transgenic line (**g**), the replenishment of Egfl7 in LECs under *Tg(lyve1b:egfl7-p2A-GFP)* is able to re-form the BLECs bilateral loop in the *egfl7* mutant (**i**). And overexpression of Egfl7 in WT lymphatics shows the BLECs are unaltered (**f**, **h**). **j** The statistics show the number of BLECs per double loops in WT and *egfl7* mutant under *Tg(lyve1b: GFP)* and *Tg(lyve1b:egfl7-p2A-GFP)* transgenic lines (*n* = 24 embryos, 2-way ANOVA. Box plots show the five-number summary of a set of data: including the minimum score (shown at the end of the lower whisker), first (lower) quartile, median, third (upper) quartile, and maximum score (shown at the end of the upper whisker)). Scale bar, 50 μm.

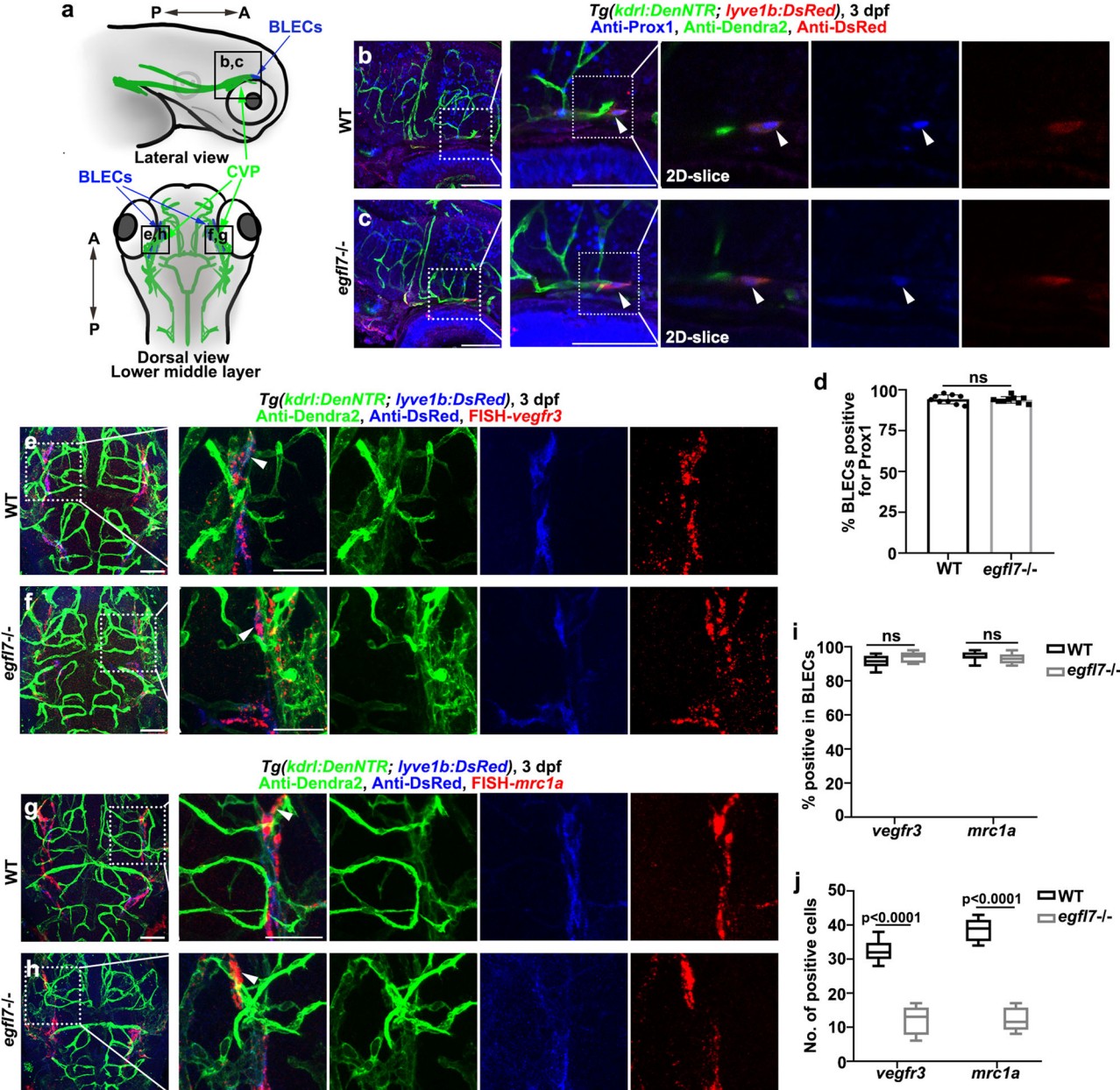

**Fig. 4 | *egfl7* is not required for the specification of BLECs progenitors sprout from CVP. a** Schematic diagram showing the lateral view and the lower middle layer of the brain, respectively. Black frames indicate the image area of corresponding panels. CVP, choroidal vascular plexus. **b, c** Immunofluorescence staining for blood endothelial cells (Anti-Dendra2), BLECs (Anti-DsRed), and Anti-Prox1 (blue nuclei) in the WT (**b**) and *egfl7* mutant (**c**) at 3 dpf. Arrowheads of the magnified 2D single slice image point to the Prox1+ BLECs progenitors sprout from CVP. **d** The statistics show the ratio of BLECs progenitors positive for Prox1 in WT and *egfl7* mutant (*n* = 9 embryos; two-tailed unpaired *t* test; ns, no significance. Data are represented as mean ± SD). **e–h** Confocal image showing the triple labeling of FISH-*vegfr3* or FISH-*mrc1a*, anti-DsRed, and Dendra2 in *Tg(kdrl:DenNTR; lyve1b:DsRed)* transgenic background in WT (**e, g**) and *egfl7* mutant (**f, h**). Arrowheads in the magnified image shows *vegfr3* mRNA and *mrc1a* mRNA are expressed in the *lyve1b* positive and low-level *kdrl* expressing BLECs progenitors departing the CVP. **i** The statistics show the ratio of BLECs positive for *vegfr3* and *mrc1a* in WT and *egfl7* mutant (*n* = 8 embryos; 2way ANOVA multiple comparisons test; ns, no significance. Box plots show the five-number summary of a set of data: including the minimum score (shown at the end of the lower whisker), first (lower) quartile, median, third (upper) quartile, and maximum score (shown at the end of the upper whisker)). **j** The statistics show the number of *vegfr3*+ and *mrc1a* + BLECs in WT and *egfl7* mutant (*n* = 8 embryos; 2way ANOVA multiple comparisons test. Box plots show the five-number summary of a set of data: including the minimum score (shown at the end of the lower whisker), first (lower) quartile, median, third (upper) quartile, and maximum score (shown at the end of the upper whisker)). Scale bar, 50 μm.

## Overexpression of Vegfc hardly rescues BLEC formation in the *egfl7* mutant

BLEC sprouting depends on Vegfc, Vegfd, Ccbe1, and Vegfr3 signaling[15]. The BLECs populating in the loop-like structures were significantly reduced in the *vegfc*-/- and *vegfd*-/- single mutants and completely absent in *vegfc*-/-*vegfd*-/- double mutant[16]. To test whether the roles of Egfl7 in BLEC proliferation and migration were dependent on classic Vegfc/Vegfr3, we injected *hsp70l: Vegfc-P2A-Venus* plasmid to generate an ectopic, mosaic expression of Vegfc in embryos, using *hsp70l: Venus* plasmid as control (Fig. 6a). We firstly analyzed trunk lymphangiogenesis to ensure the Vegfc overexpression. The trunk LECs firstly sprout from the PCV dependent on Vegfc/vegfr3 signaling, and they are highly proliferative as well as migratory when colonizing the HM to form PLs[11,38–40]. After heat-shock, the ectopic expression of

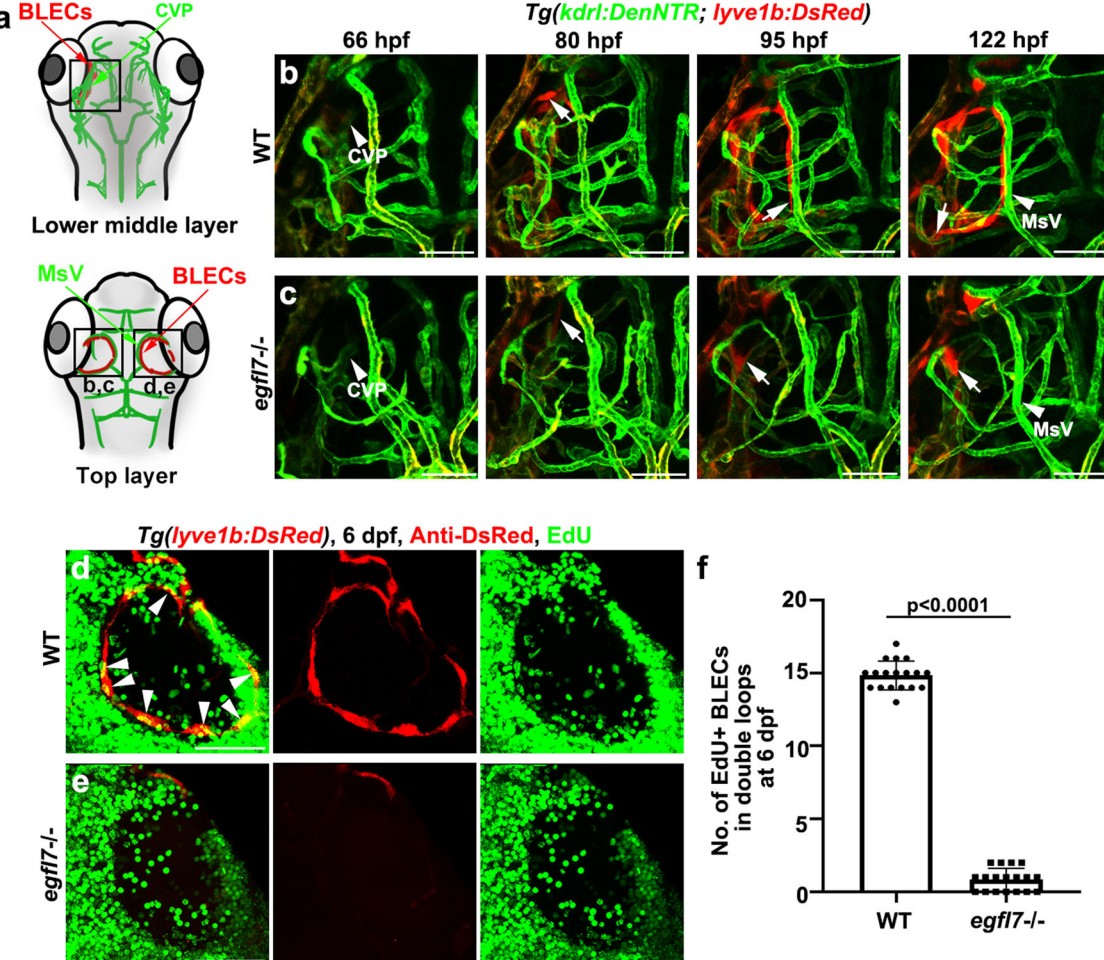

**Fig. 5 | *egfl7* is essential for the migration and proliferation of BLECs.**
**a** Schematic diagram showing the lower middle layer and top layer of the vessels in the brain, respectively. Black frames indicate the image area of corresponding panels. CVP (choroidal vascular plexus) is observed in the lower middle layer; MsV (mesencephalic vein) is in the top layer. **b** Time-lapse image of *Tg(lyve1b: DsRed; kdrl: DenNTR)* from 66 hpf to 122 hpf show the BLECs (arrows) sprout from CVP (arrowhead at 66 hpf) and migrate along the MsV (arrowhead) to form the lymphatic loop in WT at 122 hpf. $n = 15$ embryos. **c** The BLECs (arrows) of *egfl7* mutant shows a delayed sprouting from CVP (arrowhead at 66 hpf), fail to migrate along the MsV (arrowhead), and is unable to form the loop at 122 hpf. $n = 15$ embryos. **d–f** EdU staining shows the proliferation of BLECs in WT and *egfl7* mutant at 6 dpf, the EdU is injected at 56 hpf. Note the EdU+ BLECs is significantly decreased in the mutant (**d**, **e**). Arrowheads indicate the EdU+ BLECs in WT. The statistics show the number of EdU+ BLECs in double loops at 6 dpf in WT and *egfl7* mutant (**f**, $n = 18$ embryos; two-tailed unpaired *t*-test). Data are represented as mean ± SD. Scale bar, 50 μm.

Vegfc, but not the control Venus protein, resulted in the hyperproliferation and hyperbranching of venous-derived ECs prominently in the HM, either in WT or mutant (Fig. 6a, b, d, f, g and Supplementary Fig. 7a, b, d, f). These results ensure that the Vegfc overexpression experiment is working. Next, we overexpress Vegfc in the brain. In contrast to the Venus protein, the ectopic expression of Vegfc was ineffective to the BLEC deficiency in the *egfl7* mutant (Fig. 6c, e, h and Supplementary Fig. 7c, e, g). These observations suggest that the roles of Egfl7 in BLEC development doesn't seem to act upstream of Vegfc.

### Egfl7 promotes BLEC formation through integrin αvβ3 under the regulation of ILK

Egfl7 is deposited in the ECM upon secretion from endothelial cells, and it has been reported as an integrin ligand strongly associated with fibronectin[22]. To understand potential mechanisms involved in the *egfl7*-regulated BLEC formation, integrins previously reported to be expressed by LECs to regulate lymphatic development were analyzed. Among at least 24 unique integrins subunits, integrin α5β1 attached to the ECM to trigger VEGFR3 phosphorylation and LEC proliferation[41]; integrin α4β1 plays a direct role in regulating lymphangiogenesis[42]. So, we first carried out FISH and antibody staining and showed the expression of integrins including *itga4*, *itga5*, *itgb1a*, and *itgb1b* in BLECs (Supplementary Fig. 8a–f). To investigate the roles of integrin α5β1 in BLEC development, we applied an integrin α5β1 inhibitor[43], ATN-161, to treat the larvae from 54 hpf to 5 dpf. The BLEC formation remained normal after ATN-161 treatment, with no notable differences in contrast to the control group (Supplementary Fig. 8g, h, k). We then examined BLEC formation using a previously reported *itga4^cas010* mutant[44]. Additionally, when IgG647 was injected into the ventricle at 5 dpf, normal BLEC uptake was observed in the *itga4^cas010* mutant (Supplementary Fig. 8i–k). Thus, these findings suggest that integrin α5β1 and α4β1 are not involved in the Egfl7-driven BLEC formation.

Recent studies have shown that out of all the integrins, αvβ3 is a specific receptor of EGFL7[24]. The expression of *itgb3b*, *itgb3a*, and *itgav* was detected in the sprouting BLECs and CVPs from 54 hpf to 4 dpf (Fig. 7a–h). Additionally, we performed co-immunoprecipitations in HEK293 T cells to confirm the physical association of Egfl7 with Itgav and Itgb3b (Fig. 7i). To examine the impact of integrin αvβ3 on BLEC formation, we treated the *Tg(lyve1b: DsRed; kdrl: DenNTR)* transgenic line with an inhibitor of Integrin αvβ3, Cilengitide[45,46], from 54 hpf to 5 dpf. This inhibitor blocked the formation of BLEC bilateral loop-like structures (Fig. 7j–l). Additionally, we used an alternative knockdown

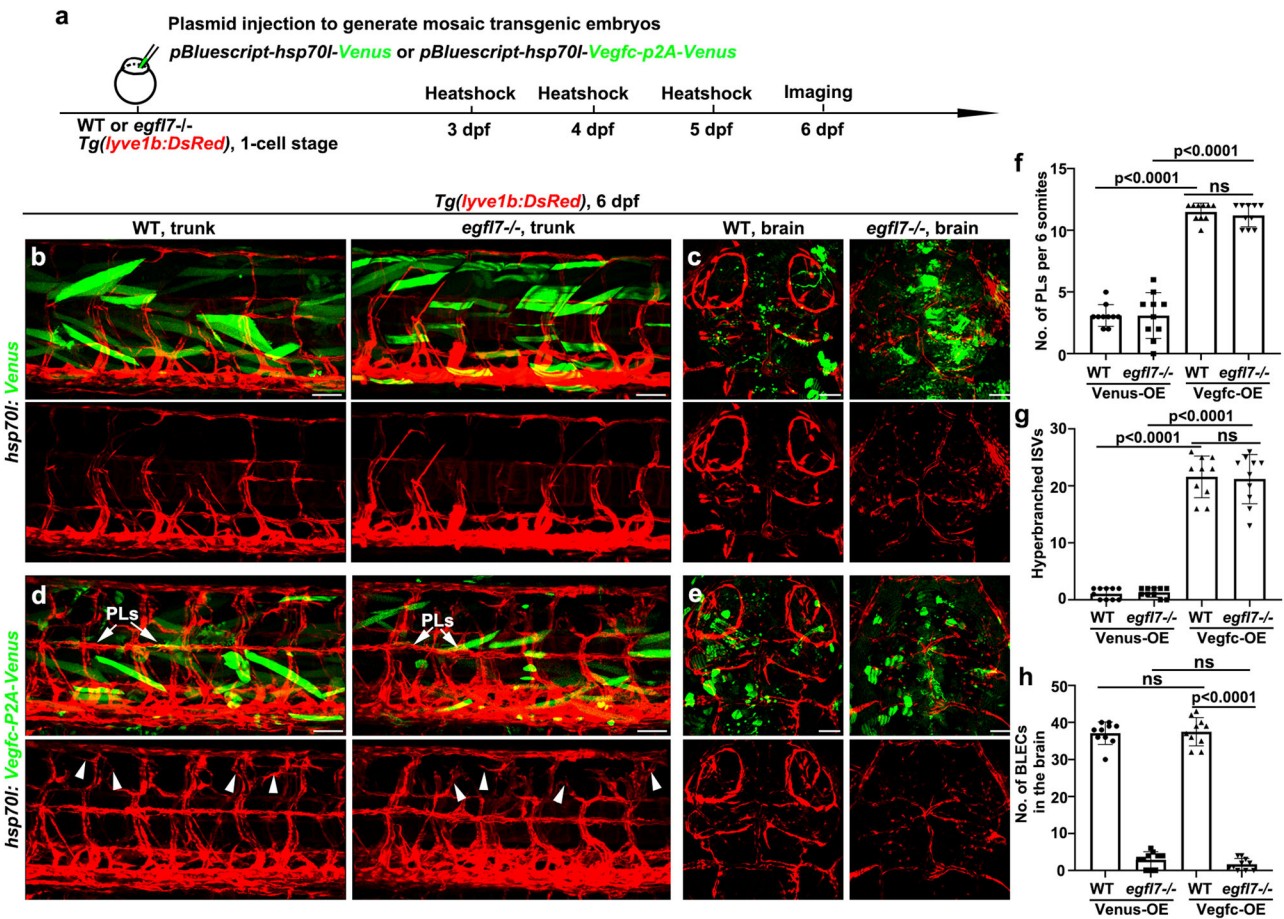

**Fig. 6 | Vegfc overexpression fails to rescue the devoid of BLECs caused by *egfl7* mutation. a** Schematic diagram showing the experimental design for detecting the development of LECs after the ectopic, mosaic expression of Vegfc and Venus. Illustrations of the method, transgenic lines, plasmids, and time points of heat shock, and imaging. **b, d** Compared to the heatshock of Venus, overexpression of Vegfc, causing a hyperbranching in adjacent PLs (arrows) and hyperbranched ISVs in the horizontal myoseptum (arrowheads) of WT (**b**) and *egfl7* mutant (**d**). **f, g** The statistics shows the number of PLs per 6 somites (**f**) and the number of hyper-branched ISVs (**g**) after ectopic expression of Vegfc and Venus in the trunk (*n* = 10 embryos; two-tailed unpaired *t*-test). **c, e** Compared to the heatshock of Venus, overexpression of Vegfc in the brain does not rescue the devoid of BLECs in the *egfl7* mutant. **h** Quantification of the number of BLECs in the brain after ectopic expression of Vegfc and Venus in the brain (*n* = 10 embryos; two-tailed unpaired *t*-test). Data are represented as mean ± SD. Scale bar, 50 μm.

approach and took advantage of the CRISPRi technology (dCas9-KRAB) to inhibit *itgb3b* transcript elongation[47]. Compared with the uninjected group (Control), the relative *itgb3b* expression levels quantified by qPCR on pools of embryos were decreased in the dCas9-KRAB mRNA and *itgb3b* gRNA co-injected group (Fig. 7o). Moreover, injected dCas9-KRAB mRNA and *itgb3b* gRNAs could inhibit BLECs loop-like structure formation in the brain (Fig. 7m–p). These results indicate that Egfl7 interacts with integrin αvβ3 to regulate the BLEC development.

It has been reported that the integrin-linked kinase (ILK) interacts with integrin in quiescent LECs to prevent non-physiological hyper-activation of VEGFR3 signaling. However, when integrin binds to the ECM, mechanical stimulation disrupts the association of ILK and integrin, releasing the integrin to interact with VEGFR3 and induce VEGFR3 tyrosine phosphorylation[48]. The expression of *ilk* could be detected in the BLECs at 4 dpf. (Fig. 8a). To explore the roles of ILK in the Egfl7-involved BLEC formation, we incubated the wild-type and *egfl7*<sup>cq180</sup> with cpd22, an inhibitor of ILK, from 54 hpf to 6 dpf. The cpd22 partially rescued the BLEC development in the *egfl7* mutant (Fig. 8b–g). In addition, we generated an *ilk* mutant, but it caused severe myocardial dysfunction and deformation, as previously described (Supplementary Fig. 9)[49]. Therefore, we used the *ilk* het-erozygous mutant for the rescue experiment. The results showed that the absence of loop-like structures in the *egfl7* mutant was partially

rescued when it was crossed with the *ilk* heterozygous mutant (Fig. 8h–k). These results demonstrate that even without Egfl7, downregulation of ILK could release a portion of integrin αvβ3 to enhance the phosphorylation of Vegfr3, thus promoting lymphangio-genesis (Supplementary Fig. 10).

## Discussion

From these findings, we proposed that in the WT, Egfl7 is deposited in the ECM upon secretion from BECs and LECs. Then, Egfl7 activates Integrin αvβ3, which disrupts the Integrin-ILK association and increa-ses the Vegfr3 phosphorylation, in turn promotes BLEC proliferation. In the *egfl7* mutant, ILK interacts with Integrin αvβ3, and Vegfr3-Integrin αvβ3 dissociates, thus reducing Vegfr3 phosphorylation and inhibiting BLEC proliferation (Supplementary Fig. 10). These observa-tions reveal that Egfl7-Integrin αvβ3 mediated ILK-Vegfr3 signaling plays a crucial role in regulating the migration and proliferation of BLECs.

It has been noted that Egfl7 plays a role in vascular repair, CNS inflammation, and cancer angiogenesis. We investigated its relevance in ischemic stroke by examining the expression of Egfl7 during brain vascular injury. We found that Egfl7 is activated not only in both ingrown lymphatic vessels and regenerating blood vessels but also highly expressed in all uninjured BLECs and blood vessels, indicating its significance role in BLECs formation and blood vessel regeneration.

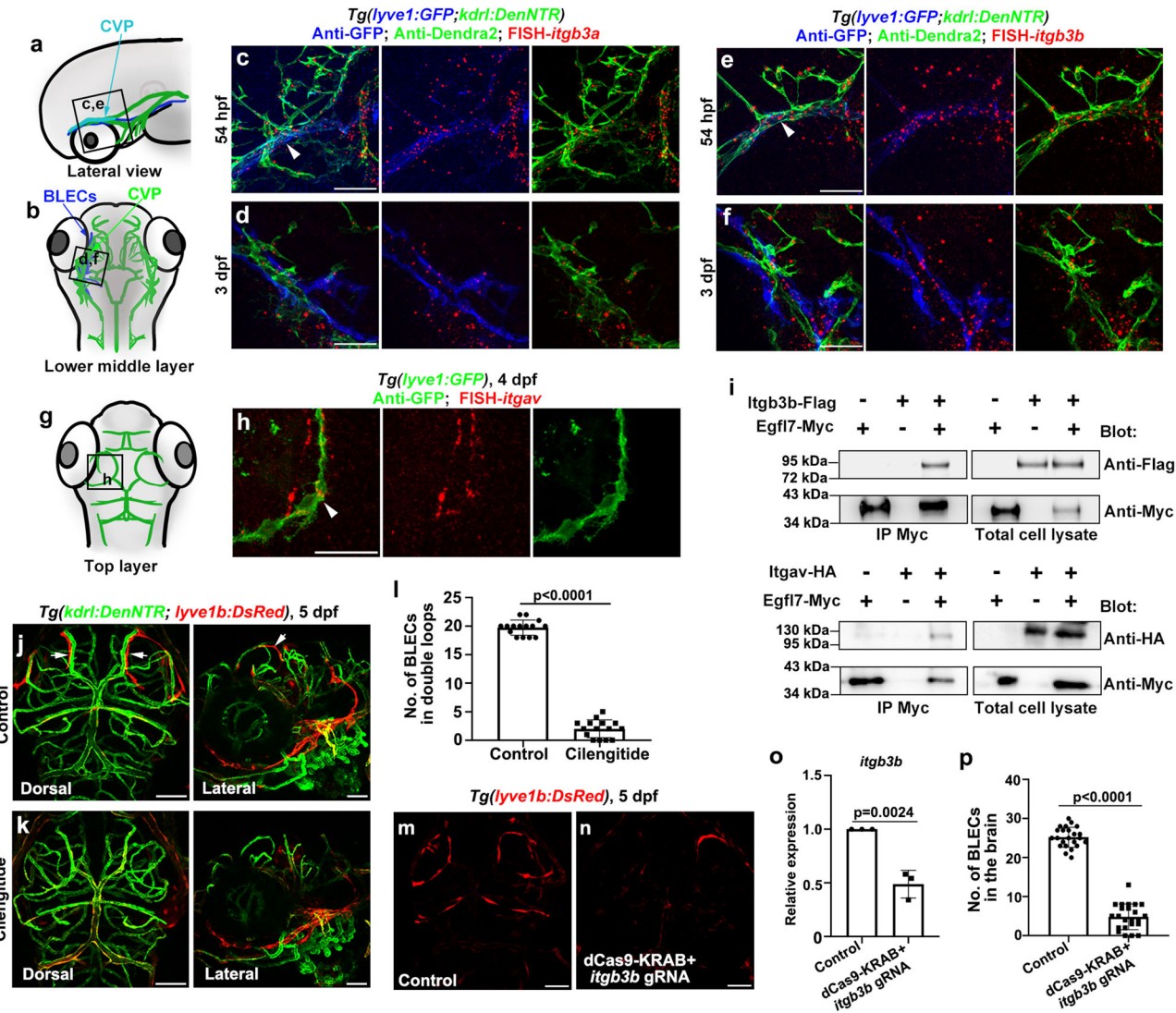

**Fig. 7 | Egfl7 interacts with integrin αvβ3 to regulate BLECs formation.**
**a**, **b**, **g** Schematic diagram showing the lateral view, the lower middle layer, and the top layer of the vessels in the dorsal view, respectively. Black frames indicate the orientation and the image area of corresponding panels. **c**–**f** In *Tg(lyve1b: GFP; kdrl: DenNTR)* transgenic background, FISH and antibody staining show *itgb3a* and *itgb3b* are expressed in CVPs at 54 hpf (**c**, n = 20/21 embryos, **e** n = 20/20 embryos, arrowheads), and expressed in BVs and departed BLECs at 3 dpf (**d**, n = 20/20 embryos, **f**, n = 19/20 embryos). **h** Double labeling of FISH-*itgav* and anti-GFP in *Tg(lyve1b: GFP)* transgenic background at 4 dpf, arrowheads points the *itgav* expressed in BLECs. n = 15/16 embryos. **i** HEK293T cells were transfected with Egfl7-Myc and Itgb3b-Flag or Itgav-HA. Egfl7 was immunoprecipitated using anti-Myc antibody and the immunoprecipitants were analyzed for the presence of Itgav-HA or Itgb3b-Flag by Western blot. **j**–**l** Inhibition of integrin αvβ3 by Cilengitide treatment from 54 hpf to 5 dpf phenocopy the devoid of the lymphatic loop in the brain (**k**). Arrows point to the BLECs in the control group (**j**). The statistics show the number of BLECs in the double loops after Cilengitide treatment (**l**, n = 15 embryos; two-tailed unpaired *t*-test. Data are represented as mean ± SD). **m**–**p** The *itgb3b* transcript elongation inhibition causes BLECs reduction in the brain. Dorsal view of BLECs loop-like structure of uninjected siblings and samples co-injected with dCas9-KRAB mRNA and *itgb3b* gRNAs (**m**, **n**). The co-injection of gRNA and CRISPRi results in decreased expression of *itgb3b* and the number of BLECs (**o**, n = 3 replicants; **p**, n = 23 embryos; two-tailed unpaired *t*-test. Data are represented as mean ± SD). Scale bar, 50 μm.

In *egfl7 cq180*, we observed a lack of BLECs but normal BV development, and the reduced BLECs resulted in the failure of lymphatics response to blood vascular injury[20], leading to brain edema and death. Despite the absence of BLECs, the *egfl7 cq180* can mature into adulthood and appear normal and fertile under bright field imaging. However, it exhibits abnormal meningeal blood vasculature[16]. Further studies are needed to determine if the absence of BLECs affects the behavior and neurons of adult zebrafish.

The *egfl7 cq180*, is similar with *egfl7 s981*, does not show any apparent vascular abnormalities, which can be explained by the activation of compensatory genes-Emilin3. It contains the EMI domain, can regulate vascular elastogenesis[29]. However, the upregulation of Emilin3 cannot replace the EGF domain, and therefore, it may not be able to rescue the

*egfl7 cq180* on BLECs formation. Further investigation is required to determine the functional role of different Egfl7 protein domains and ascertain whether the lack of BLECs is due to the absence of the EGF domain.

Integrin, as another receptor of lymphatic endothelial cells, has been reported in lymphangiogenesis. Integrin is a transmembrane receptor, which binds to extracellular matrix components, and is essential for "outside-in" and "inside-out" signaling of the cell, thereby transducing mechanical stimulations[8]. Surprisingly, our research has found that integrin α4β1 and α5β1 are not essential in the Egfl7-driven lymphatic sprouting in the brain. Instead, integrin αvβ3, which specifically binds to Egfl7[24], plays a crucial role in brain lymphangiogenesis. We have observed that inhibiting integrin αvβ3 with Cilengitide blocks

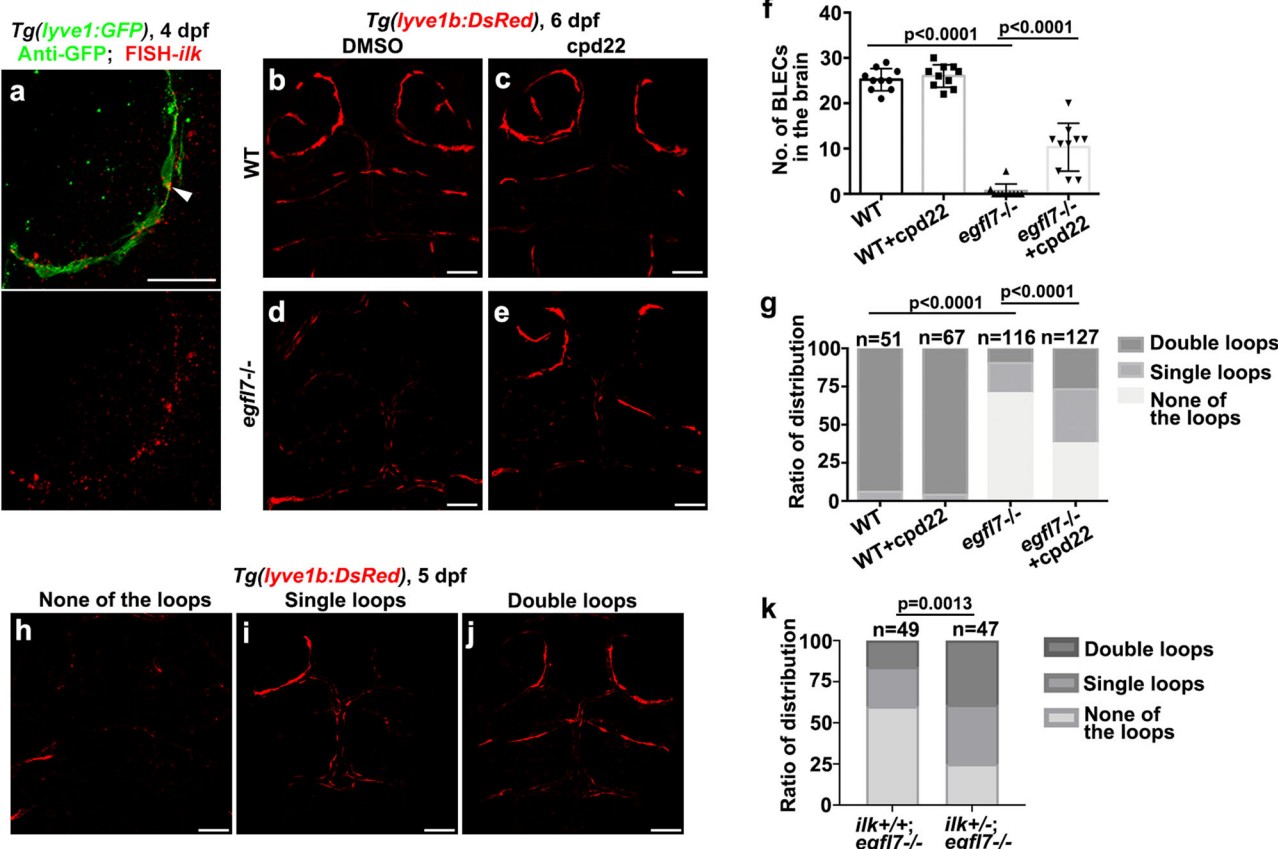

**Fig. 8 | Egfl7-Integrin αvβ3 plays a role in BLECs formation through ILK pathway. a** FISH and antibody staining shows *ilk* is expressed in GFP+ BLECs at 4 dpf (arrowheads). *n* = 17/18 embryos. **b–g** Inhibition of ILK by cpd22 can partially rescue the absence of BLECs in the *egfl7* mutant at 6 dpf (**b–e**). Quantification of the number of BLECs in the double loops of brain in different treatment groups (**f**, *n* = 10 embryos, two-tailed unpaired *t* test. Data are represented as mean ± SD). The statistics show the percentage of embryos that have double lymphatic loops, single loops, and none of the loops in the brain (**g**, WT, *n* = 51 embryos, WT + cpd22, *n* = 67 embryos, *egfl7*-/-, *n* = 116 embryos, *egfl7*-/- + cpd22, *n* = 127 embryos, χ² test).).

**h–k** An *egfl7* mutant was crossed with an *ilk* heterozygote to generate two types of larvae: *ilk*+/+; *egfl7*-/- and *ilk*+/-; *egfl7*-/-, which were then studied for BLECs loop-structures formation. The larvae were classified into three categories based on their phenotype: Double loops, single loops, and none of the loops (**h–j**). The results showed that compared to *ilk*+/+; *egfl7*-/-, the *ilk*+/-; *egfl7*-/- larvae had a decreased percentage of the none of the loops phenotype, and a partially rescued percentage of double loops and single loops phenotype (**k**, *ilk*+/+; *egfl7*-/-, *n* = 49 embryos, *ilk*+/-; *egfl7*-/-, *n* = 47 embryos, χ2 test). Scale bar, 50 µm.

brain lymphangiogenesis in zebrafish, mimicking the phenotype of the *egfl7 cq180*. Furthermore, pharmacological inhibition of ILK has rescued the absence of BLECs caused by *egfl7* mutation. These findings suggest that EGFL7 regulates BLECs formation by binding to integrin αvβ3, while Ilk interacts with integrins to facilitate mechanically regulated VEGFR3 signaling. Our future research will involve generating the *itgav*, *itgb3a*, and *itgb3b* mutant by Cas9 to mimic the Cilengitide inhibition experiment and the conditional knock out of *ilk* mutant to rescue the *egfl7 cq180*. These results also suggest that compared with trunk lymphangiogenesis depending on Vegfc-Ccbe1-Vegfr3, brain lymphatics development is primarily depending on Egfl7-Integrin αvβ3 signaling.

The specific functions of Egfl7 in the endothelial system have been a topic of controversy due to varying phenotypes observed in different knockout (KO) alleles of EGFL7 in mice. Some studies have reported angiogenesis deficits in Egfl7 gene-trap and lacZ knock-in mice[50], while *Egfl7ᐃ/ᐃ* mice appeared to be normal[28]. Although the earlier study attributed vascular phenotypes of *Egfl7* KO to the loss of *miR-126*, the *Egfl7-/-* that maintains *miR-126* expression in another study demonstrates *Egfl7* is crucial for placental vascularization and embryonic growth[51], suggesting a specific role for *Egfl7* in vascular development. Zebrafish BLECs correspond to the mouse leptomeningeal LECs (LLECs)[17]. And Egfl7 was increased in CNS vasculature of mice with experimental

autoimmune encephalomyelitis (EAE). Egfl7-KO or EC-restricted Egfl7-KO mice showed earlier onset of EAE[33]. Except that, the EAE could induce VEGFR3-dependent lymphangiogenesis[52]. These studies provide indirect evidence that brain lymphangiogenesis of mice also correlated with *Egfl7*. However, due to the varying phenotypes observed in different *Egfl7*-/- mice, further investigations are needed to determine the appropriate *Egfl7* mutant of mice for researching the molecular mechanisms of *Egfl7* on brain lymphangiogenesis.

Our research has significant implications for the development of brain lymphatics. Firstly, we have discovered that Egfl7 is the primary regulator for brain lymphatic sprouting in zebrafish, rather than Vegfc. Secondly, we have found that *egfl7* may not be necessary for angiogenesis, but it plays a crucial role in lymphatic development, particularly in brain lymphangiogenesis. Thirdly, our study provides evidence that the development of BLECs is regulated by specific lymphatic growth factors different from classic signaling involved in trunk lymphangiogenesis. Egfl7 is essential for brain lymphatics development, and its implications are relevant for both normal and pathological conditions. Disruption of brain lymphatic function has been linked to neurodegenerative diseases such as Alzheimer's and Parkinson's disease. Further research on Egfl7 could uncover new therapeutic targets and improve outcomes for patients with neurodegenerative diseases, making it an exciting area of study in the field of lymphatics biology.

## Methods

### Ethical Approval

The zebrafish facility and study were approved by the Institutional Review Board of Southwest University (Chongqing, China). Zebrafish were maintained in accordance with the Guidelines of Experimental Animal Welfare from Ministry of Science and Technology of People's Republic of China (2006). The protocols used for animal experimentation were approved by the Institutional Animal Care and Use Committee protocols from Southwest University (IACUC, 2007). The IACUC number is: IACUC-20231201-01.

### Zebrafish strains

The *Tg(kdrl:DenNTR)*[cq10 20], *Tg(lyve1b:GFP)*[cq8621], *Tg(lyve1b:DsRed)*[cq2720], *Tg(kdrl:CFP-NTR)*[cq6220], *Tg(prox1:KalTA4;UAS:TagRFP)*[nim5Tg34], *Tg(egfl7:YFP)*[cq181], *Tg(kdrl:egfl7-p2A-GFP)*[cq182], *Tg(lyve1b:egfl7-p2A-GFP)*[cq183], *egfl7*[cq180], and *itga4*[cas010 44] (a gift from Weijun Pan), *ilk*[cq187] mutant were used in this study. Embryos were treated with 0.003% 1-phenyl-2-thiourea (PTU, Sigma) to inhibit pigment formation. The sex of zebrafish aged from 1 dpf to 12 dpf was unknown. For adult zebrafish imaging, both males and females were used for this experiment. All the strains are used as stable, germline transgenic lines in this study. Administering general anaesthesia to zebrafish using pharmaceutical-grade buffered Tricaine (MS222, Sigma).

### Molecular cloning

For *egfl7* promoter, from −3.5 kb to −1.5 kb genomic sequence upstream from the transcription start site combined with exon 1 and a 0.94-kb enhancer sequence located on intron 1 of the *egfl7* gene was amplified from genomic DNA. The two pair primers: Egfl7 (−3.5_−1.5)-FW: 5′-CTTCCCCCACTGATAACACATAC-3′; egfl7 (−3.5_−1.5)-RE: 5′- AGGCATGCAGACATGCTCAAGA-3′; Egfl7 (EP)-FW: 5′- GAGGGCCGAGGCGGGAGTGTTTATG-3′; egfl7 (EP)-RE: 5′- CGCTATGCTAAAATCCAGTTGGGCA-3′ were used. Then the two fragments were combined by overlap PCR and subcloned into the *pBluescript-kdrl* vectors between the *Xho*I and *EcoR*I sites to replace the *kdrl* promoter. To construct the *egfl7:YFP* plasmid, *YFP* fragment in *pGEMT-YFP* was cleaved by *Spe*I and *Xma*I, and subcloned downstream of *egfl7* promoter to generate the *pBluescript -egfl7:YFP* construct.

For constructing the *pBluescript-kdrl-egfl7-p2A-GFP* plasmid and *pT2KXIGD-lyve1b-egfl7-p2A-GFP* plasmid, we first amplified the full-length *egfl7* cDNA by the primers egfl7-CDS-FW: 5′-ACCGGTATGTACACAGCGCTTCTGCTC-3′ and egfl7-CDS-RE: 5′-GTTTTCCTGACAGCCACAGGCTC-3′, then the *egfl7* sequence overlapped with the *p2A-GFP* sequence was inserted between the *Age*I and *Not*I sites in the *kdrl:Dendra2* plasmid to replace the *Dendra2* fragment. This was done to construct the *pBluescript-kdrl:egfl7-p2A-GFP*. The *pT2KXIGD-lyve1b-egfl7-p2A-GFP* plasmid was created using the In-Fusion cloning method (Clontech, #639619). Initially, we designed PCR primers to amplify the *egfl7-p2A-GFP* fragment with an added 5′ tail, creating 15 bp overhangs for annealing to the vector. The In-Fusion primers used were 5′-AATCCAAGGGATCCA-3′ and 5′-TCTGGATCATCATCG-3′. We then linearized the *pT2KXIGD-lyve1b: GFP* plasmid between the *Age*I and *Cla*I sites to obtain the vector. The *pT2KXIGD-lyve1b-GFP* plasmid was kindly provided by Philip S. Crosier. Finally, we performed the In-Fusion reaction to ensure the *egfl7-p2A-GFP* fragment was inserted into the vectors and validated the fusion points for any mutations via sequencing.

For constructing *pcDNA3.1-egfl7*[WT]*-Myc* and *pcDNA3.1-egfl7mut-Myc*, *pcDNA3.1-Itgb3a-Flag*, and *pcDNA3.1-itgav-HA* plasmids for cell transfection. cDNA was generated using Omniscript reverse transcriptase (Qiagen, #205113). cDNAs encoding the Egfl7[WT] (831 bp/277 aa), Egfl7[mut] (213 bp/71 aa) proteins were PCR-amplified using cDNA from WT or *egfl7*-/- as template, followed by ligated into the mammalian expression vector *pcDNA3.1 Myc-tag* by In-Fusion cloning, the Myc-tag was fused to the C-terminal of the cDNAs. CDS encoding full length of *Itgav* and *itgb3a* protein were PCR-amplified. These PCR fragments were ligated into the mammalian expression vector *pcDNA3.1 HA-tag* or *pcDNA3.1 Flag-tag* by In-fusion method. All constructs were verified by sequencing.

### Generation of transgenic line

For generation of *Tg(egfl7:YFP)*[cq181] transgenic line. Co-injection of *I-Sce*I (NEB) with *pBluescript -egfl7:YFP* construct flanked by the *I-Sce*I restriction sites into AB genetic background at the one-cell stage for transgenesis. The embryos were screened for transient expression of YFP in endothelium after injection for 3 days by using the Leica epifluorescence microscope. The vasculature of injected embryos showing specific expression of YFP was raised to adulthood and crossed with AB fish to get the germline transmissions. In the stable transgenic line, all the endothelial cells including lymphatic endothelial cells and blood vascular endothelium express the fluorescent protein YFP.

To rescue Egfl7 in BVs and LECs of *egfl7*[cq180], two stable transgenic lines were generated - *Tg(kdrl:egfl7-p2A-GFP)*[cq182] and *Tg(lyve1b:egfl7-p2A-GFP)*[cq183], respectively. The *Tg(kdrl:egfl7-p2A-GFP)* was created using the pBluescript vector. The constructs flanked by the *I-Sce*I restriction sites were co-injected along with *I-Sce*I (NEB) into the *egfl7* mutant at the one-cell stage for transgenesis. After injection, the embryos were screened for transient expression of GFP under a Leica epifluorescence microscope. The embryos that exhibited specific expression of GFP in the vasculature were raised to adulthood for identification of founder fish with germline integration. To generate *Tg(lyve1b:egfl7-p2A-GFP)*, the plasmid *pT2KXIGD-lyve1b-egfl7-p2A-GFP* (25 ng/µl) with transposase mRNA (25 ng/µl) were co-injected into one-cell stage of *egfl7* mutant embryos and the progeny screened for germline transmission. In the stable transgenic line, all lymphatic endothelial cells express the fluorescent protein-GFP.

### Generation of genetic mutants and genotyping

The CRISPR-Cas9 technique was used to generate the mutants. The gRNA target sequence of *egfl7* and *ilk* showed in the Figures, and the gRNAs were synthesized as described[53,54]. Co-injection of Cas9 protein and gRNAs into one-cell stage of AB genetic background, the genomic region flanking gRNA target site was amplified with the pairs of gene-specific primers including: egfl7 ID-F: 5′-GGAAGTCAACAGCACCTTGAGGG-3′, egfl7-ID-R: 5′-CCTGCCTATAAGAAACCTTGTAG-3′; ilk-ID-F: 5′-GAACAAGATCAACGAGAACC-3′; ilk-ID-R: 5′-TATCGCAATAACTAGCAGCC-3′; itga4-ID-F: 5′-GGTGCTCTGACTGATGACGA-3′, itga4-ID-R: 5′-ATAGGTACAATCCGCGCAAC-3′[44], and used these primers to do the PCR and sequenced for validation. The validated embryos were raised to adults (F0). The F0 fishes were screened to identify the founders whose progenies carried indels in these genes. The F1 embryos from the identified F0 were raised to adult, and every F1 adult was identified by genotyping and sequence.

### CRISPR interference

The *pXT7-dCas9-KRAB* plasmid[47] was linearized by BamH1 restriction enzymes to serve as templates to synthesize stable mRNAs. Synthesis and purification of dCas9-KRAB mRNA were described previously. The two *itgb3b* gRNAs were designed to target the non-template strand of the 5′ untranslated region (UTR) and exon 1 of *itgb3b* to block transcriptional elongation. The target sites are 5′-GCTGTTCTTCTCTACTTCAC-PAM-3′ and 5′-GTAAACCCATAAGCTGAAGAGTT-PAM-3′, and the gRNAs were synthesized as described above. mRNA of dCas9-KRAB (400 ng/µl) and two *itgb3b*-gRNAs (200 ng/µl) were co-injected into 1-cell stage zebrafish embryos.

### Real-time quantitative PCR

Total RNA of *egfl7*[cq180], wildtype, and embryos that injected with dCas9-KRAB mRNA and *itgb3b* gRNAs were prepared using the TRIzol reagent (Invitrogen, #15596026). cDNA was generated using Omniscript reverse

transcriptase (Qiagen, #205113) with specific stem–loop primers for miRNA. Real-time quantitative PCR was performed using SYBR Green (Roche, #04913914001). The relative miRNA amount was calculated with the $\Delta\Delta Ct$ method and normalized with internal control U6 snRNA as previously described[55]. The *egfl7* mRNA expression was normalized by transcriptions of *β-actin*. RT primer for miR-126: 5′-TGGAGCGAC CGTGTCGTGGAGTCGGCTAATGGTCGCTCCATGCAC-3′ ; Primers for miR-126a/b Real-Time PCR: 5′-GACACTCCAGCAGCGTCGTACCGTGAGT AATA-3′ and 5′-ATAGAGCGGTGTCGTGGAGTCGGCTAATGGTC-3′; Primers for u6 Real-Time PCR: 5′-ACTAAAATTGGAACGATACAGAGA-3′ and 5′-AAAGATGGAACGCTTCACG-3′. Primers for *egfl7 qPCR ex1*: 5′- ATGTG ACCTGCACACGTCAG-3′ and 5′- CACACGACGACTCCAGACAT-3′; Primers for *egfl7 qPCR ex2*: 5′-ATGTGCCAAAACCACCACATG-3′ and 5′-GA GCCTCCGTTTGCACAAGACT-3′; Primers for *itgb3b qPCR ex2*: 5′-GATGT TGGACTAGGTTCTAACGT-3′ and 5′-CTGAGGCCTTGTCACTGAGATC-3′; *β-actin*-qPCR primers: 5′-CGTCTGGATCTAGCTGGTCGTGA-3′ and 5′-CAATTTCTCTTTCGGCTGTGGTG-3′.

## Cell transfection and Western blot

HEK293T cells were purchased form Cell Bank, Chinese Academy of Sciences (Shanghai, China), the cells were maintained in DMEM medium (Gibco, #11965092) (containing 10% FBS (Gibco) and 1% penicillin & streptomycin) and cultured at 37 °C, 5% (v/v) $CO_2$. Transfection was performed with Lipofectamine™ 3000 Transfection Reagent (Invitrogen, #L3000-001) and tested according to the protocol provided by the manufacturer.

Cells were lysed with lysis buffer (150 mM NaCl, 50 mM Tris-HCl, 1.0% Triton X-100, pH 8.0) containing PMSF (1 mM). Western Blot followed the standard protocol. Briefly, the protein was separated on 12% polyacrylamide gels (Bio-Rad) and transferred in a PVDF membrane. The membranes were blocked with 5% milk and incubated with the following primary antibodies: Mouse Anti-Flag (1:5000, Sigma, #F1804), Mouse Anti-HA (1:5000, Covance, #E11FF01244), Rabbit Anti-Myc (1:5000, Abcam, #ab9106). After 4 °C overnight incubating with primary antibodies, the membrane was washed 5 times with blotting buffer and incubated with appropriate horseradish peroxidase-conjugated secondary antibodies: Goat anti-mouse IgG HRP (1:5000, CST, #91196), Goat Anti-Rabbit IgG HRP (1:5000, CST, #7074) for 2 h at room temperature. After washing the membrane, SuperSignal West Pico Chemiluminescent Substrate (#34577, Thermo Fisher Scientific,) was used to visualize by chemiluminescence.

## Co-immunoprecipitation

For Egfl7-Itgb3b and Egfl7-Itgav coimmunoprecipitations, 5 µg *pcDNA3.1-Egfl7-myc*, *pcDNA3.1-Itgb3b-Flag*, *pcDNA3.1-itgav-HA* plasmids and different combinations as indicate on demand were transfected into HEK293T cells. The cells were collected after 36 h of incubation and the total cell lysates were prepared using lysis buffer. Take 1/3 of total cell lysates as input and detected the protein level by Western blot. The remaining lysates were subjected to immunoprecipitation with Rabbit anti-Myc (1:5000, Abcam, #ab9106) using Pure Proteome™ Protein A/G Mix Magnetic Beads (Millipore, #LSKMAGAG02) according to the manufacturer's instructions. Coimmunoprecipitated proteins were eluted in 2× SDS sample buffer (125 mM Tris, pH6.8, 20% glycerol, 0.02% bromophenol blue, 2% b-mercaptoethanol, and 4% SDS), separated on 12% SDS-PAGE gels, and analyzed by immunoblotting using anti-HA (Covance), anti-Myc (Abcam), or anti-Flag antibodies (Sigma).

## Whole-mount in situ hybridizations and immunofluorescence staining

The zebrafish embryos at indicated stages were fixed in 4% PFA for in situ hybridizations[56]. To avoid of off-target hybridization to tissues expressing transgenes, the digoxigenin-labeled antisense RNA probe *egfl7* was synthesized from PCR templates. Then the whole-mount

in situ hybridizations was carried out as previously described. Images were snapped using a SteREO Discovery V20 (Carl Zeiss) microscope equipped Zen 2011 software.

The embryos for whole mount immunofluorescence staining were fixed in PEM and 2% FA at 4 °C overnight[57]. Then incubated the following primary antibodies including: Anti-PCNA (1:500, Sigma, #SAB2701819), Anti-DsRed (1:1000, Santa Cruz, #sc-101526), Anti-Dendra2 (1:1000, Antibody-online, #ABIN361314), Anti-Prox1 (1:500, Abcam, #ab5475). After washing with PBST several times, incubated the embryos with Alexa fluorescent-conjugated secondary antibodies (1:1000, Invitrogen) at 4 °C overnight and washed to subjected for mounting and imaging under Confocal microscope (LSM 880, Carl Zeiss). Antibody stained embryos were mounted in 1.2% low melting point agarose and imaged using ZEN2010 software equipped on an LSM 880 confocal microscope (Carl Zeiss).

## Combined FISH and antibody staining

PCR-amplified sequences of *egfl7*, *vegfr3*, *mrc1a*, *itga4*, *itga5*, *itgb1b*, *itgb3a*, *itgb3b*, *itgav*, and *ilk* were used as templates for the synthesis of antisense digoxigenin-labeled RNA probes[21].

The combination of FISH and antibody staining was performed as previously described. In short, after being fixed in 4% PFA at 4 °C overnight, the larvae older than 4 dpf were manually removed the skin under the microscope as previously described[58,59]. Then, the larvae were dehydrated in methanol at −20 °C for at least 24 h. The dehydrated samples were serially transferred into methanol in PBST (1% Triton X-100 in PBS) and pre-hybridized in the HYB buffer (50% formamide, 5×SSC, 0.1% Tween-20, 5 mg/ml torula yeast RNA, 50 mg/ml heparin) and hybridized with the digoxigenin-labeled *egfl7*, *vegfr3*, *mrc1a*, *itga4*, *itga5*, *itgb1b*, *itgb3a*, *itgb3b*, *itgav*, and *ilk* probes at 65 °C overnight. After removal of probes, the larvae were serially washed with SSCT and MABT (150 mM maleic acid, 100 mM NaCl, 0.1% Tween-20, pH 7.5), then blocked in 2% Block Reagent (Roche, #11096176001) and incubated with the Anti-digoxigenin POD antibodies (1:500, Roche) overnight. The larvae were serially washed MABT, PBST, and PBS, then incubated in TSA Plus Fluor or Cy5 Solution (Perkin Elmer, #NEL745) overnight and washed again with PBST. Afterwards, these samples were subjected to do the antibody staining.

*Tg(lyve1b:GFP; kdrl:DenNTR)* lines were subjected to antibody staining using the anti-GFP (1:1000, Santa Cruz, #sc9996) and anti-Dendra2 (1:1000, Antibody-online, #ABIN361314) primary antibodies. Then, the goat anti-mouse IgG Alexa fluor 633-conjugated (1:1000, Invitrogen, #A21052) and goat anti-rabbit IgG Alexa fluor 405-conjugated (1:1000, Invitrogen, #A31556) secondary antibodies were used to label GFP and Dendra2, respectively.

## TUNEL assay and ICV injection

Larvae were fixed in 4% PFA at 4 °C overnight, subjected to skin removal, and assayed using the In-Situ Cell Death Detection Kit, TMR Red (Roche, #04913914001) according to the manufacturer's instruction. And the positive control was generated by treating samples with DNAseI to cause DNA breakage before staining.

After embryos were mounted in 1.2% low melting agarose, a suspension of IgG-conjugated Alexa Fluor 647 (2 mg/ml, Invitrogen) was injected into the center of the optic tectum using glass capillary needles.

## EdU staining

The larvae were fixed in 4% PFA overnight at 4 °C, followed by removal of the skin. The Click-iT™ EdU Alexa Fluor™ 647 Imaging Kit (Invitrogen, #C10340) was used to assay the samples. EdU (500 µM) was injected into the CCV and ventricle of the brain in both WT and egfl7 mutant at 56 hpf or 3 dpf, as mentioned in the experiment. The samples were then fixed at the specified timepoints. The proliferation cells

were labeled with EdU, and the DsRed+ BLECs in the *Tg(lyve1b: DsRed)* transgenic line were stained with antibodies.

## Imaging

For time-lapse live imaging, the embryos were mounted in 1–1.2% low melting point agarose in the egg water with 0.003% PTU using 35 mm glass bottom dishes, and the Z-stack images of about 24 embryos were snapped every 2 h by LSM880-confocal microscope equipped with ZEN2010 software (Carl Zeiss).

For the dissected adult brain imaging, the adult was firstly anesthetized with 40 mg/L Tricaine (MS222, Sigma), then the brains were dissected and placed into cold PBS for imaging. The dissected brains were whole-mount imaged by LSM880-confocal microscope equipped with 10X Air (0.45 N.A.) and 20X water immersion (0.95 N.A.) objectives using ZEN 2010 software. The large size of the adult zebrafish brains required tile scan acquisitions that were later stitched using ZEN 2010 software.

For adult fish live imaging, the WT and *egfl7* mutant under *Tg(lyve1b: DsRed)* transgenic line (older than 6 months post fertilization, males or females) were anesthetized with 140 mg/L tricaine (MS-222) and placed upright or sidewise into a sponge slit moistened with tricaine water. The sponge containing the fish was placed in a petri dish filled with tricaine water. The fluorescence of lateral head and the dorsal head were snapped by Leica M205 FCA stereomicroscope equipped with LAS X software.

## Chemical treatment

For NTR-Mtz ablation experiment, the larvae at 3 dpf were incubated with 2 mM Metronidazole (Sigma, #M3671) dissolved in 0.2% DMSO for 2.5 h, then washed the larvae with fish water three times and recovered in embryo medium containing 0.003% PTU. After about one day, the brain vascularity of these Mtz-treated larvae showed ablation by Mtz and can be examined under the fluorescent microscope. The integrin α5β1 inhibitor-ATN-161(Selleck, #262438-43-7) and the integrin αvβ3 inhibitor-Cilengitide (MCE, #HY16141) stock solution of 10 mM in DMSO was used to prepare a working solution-50 μM and treat the embryos from 54 hpf to 5 dpf. 2.5 μM cpd22 (Calbiochem, 407331), an ilk inhibitor, dissolved in DMSO and diluted in fish water was used to treat the embryos from 54 hpf to 6 dpf.

## Mosaic ectopic expressions

For ectopic expression of Vegfc and Venus, the plasmid *pBluscript-hsp70l-Vegfc-p2A-Venus-cryaa-Venus*[20] and *pBluescript-hsp70l-Venus*[19] were used to inject into the blastomeres of *egfl7* mutants in *Tg(lyve1b: DsRed)* transgenic line at the one-cell stage[20]. Then, heat-shock was performed at the indicated time points. Heat shock was carried out at 38.5 °C for 40 min in the water bath. After heat-shock for several hours, the larvae showed green fluorescence under a microscope.

## Quantification and statistical analysis

All statistical calculations were performed using GraphPad Prism. Variance for all groups data is presented as ± SD, or Boxplot with Min to Max whiskers. All experiments were performed with embryos and adult zebrafish. The embryos were collected from in-crosses and out-crosses of several pairs of adult zebrafish. *egfl7* mutant was grown in a single tank, the other mutants and sibling larvae were grown together in a single tank. Phenotyping preceded genotyping in mutant analyses, hence analysis was genotype blinded. In the other experiments, the investigators were not blinded to group allocation during data collection and/or analysis. All experiments comparing treatment groups were carried out using randomly assigned siblings. After at least two repeated experiments (n numbers is indicated in the figure legends), data were analyzed for statistical significance using Two-way ANOVA by Sidak's multiple comparisons test, Chi-square test, and two-tailed unpaired *t*-test. A value of $p < 0.05$ was considered to be statistically

significant. No data were excluded from analyses. The exact sample size (n), p-value for each experimental group, statistical tests, and error bars (e.g. SD, SEM) were defined in the figure legends or in the Source Data.

## Reporting summary

Further information on research design is available in the Nature Portfolio Reporting Summary linked to this article.

## Data availability

All data generated and/or analysed in this study are available in the Article and the Supplementary Information. If there is potential for commercial application, we may require a payment or a completed Material Transfer Agreement. All zebrafish lines and plasmids generated in this study are available from the laboratory of L.L. (lluo@fudan.edu.cn). Source data are provided with this paper.

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

## Acknowledgements

We thank Prof. Weijun Pan for the *itga4*$^{casO10}$ mutant fish lines. This work was supported by the National Key R&D Program of China (2021YFA0805000 (L.L.)), and the National Natural Science Foundation of China (32322028 (J.C.), 32192400 (L.L.)).

## Author contributions

J.Chen and L.Luo designed the experimental strategy, analyzed data, and wrote the manuscript. J.D. generated the *egfl7* mutant and

performed the combination of FISH and antibody staining. Y.X and F.F generated the *ilk* mutant, performed CRISPRi, constructed plasmids, and performed injections. Y.Li and T.W did the chemical treatment, co-immunoprecipitation and western blot experiments. J.H, J.Cang, and L.L analyzed data and provided a discussion. J.Chen performed all the other experiments in the study.

## Competing interests

The authors declare no competing interests.
