## [Transparent Peer Review file · Nature Communications]

Epidermal growth factor-like domain 7 drives brain lymphatic endothelial cell development through integrin $\alpha\beta 3$

Corresponding Author: Professor Jingying Chen

Version 0:

Reviewer comments:

Reviewer #1

(Remarks to the Author)

This study addresses the role of the endothelial-expressed secreted factor EGFL7 in development of BLECs (aka MuLECs, FGPs), lymphatic-related perivascular meningeal cells recently discovered in zebrafish. Although a variety of knockdown studies have suggested an important role for EGFL7 in endothelial cells, knockout mice and mutant zebrafish lack obvious vascular phenotypes, at least in part because of upregulation of compensatory factors by transcriptional adaptation. In this study, the authors re-confirm lack of blood or lymphatic vessel phenotypes in EGFL7 mutant zebrafish, but they also report that these mutants fail to form BLECs and go on to carry out additional studies aimed at understanding more about how EGFL7 promotes BLEC development. Although the loss-of-BLEC phenotype is noteworthy, substantial problems with this manuscript reduce its impact and significance.

The evidence that lyve1b:dsred-positive BLECs are missing in the EGFL7 mutant appears solid, although the authors should have used additional methods to validate the actual absence of BLECs in these mutants besides loss of expression of a single transgene. The data supporting many or most of the other conclusions in this manuscript is often much less solid. This includes extensive use of selective data such as individual magnified images shown with little or no anatomical context, subjective or missing quantification of phenotypes, lack of necessary controls, and/or data that do not clearly show what the authors are concluding in their text (eg, cell autonomy data in fig. 3, specification data in Fig. 4, proliferation data in Fig. 5). Many of the descriptions of the precise methods and reagents used that are provided in the text and legends are not adequate to fully understand exactly how the experiment was done. Problems with grammar and usage throughout the manuscript sometimes also get in the way of a clear understanding of the descriptions provided. The potential significance is also overstated - the authors purport to be studying "brain lymphangiogenesis" in their title and throughout the text but BLECs are neither in the brain (they are in the meninges surrounding the brain, not in the brain proper) nor are they lymphatic vessels (they are individual separated perivascular cells on the outside of meningeal blood vessels). The authors do not look at actual zebrafish meningeal lymphatics in this study. All of this substantially detracts from the impact and significance of this manuscript.

Reviewer #2

(Remarks to the Author)

The authors describe a novel role for Egfl7 in regulating brain lymphatic endothelial cell (BLECs) development. They claim that Egfl7 is essential for sprouting and proliferation of BLECs by signaling via Integrin $\alpha\beta 3$ and mediating ILK-Vegfr3 signaling.

First of all, the requirement for Egfl7 in BLEC development is completely novel and therefore worthwhile of publication in Nature Communications. The authors show, that it is expressed in BLECs (as well as in blood vessel endothelial cells) and that loss of function impairs BLEC development, but not progenitor specification. Furthermore the newly generated egfl7 mutants survive to adulthood without gross morphological consequences or lymphedema.

After arguing the mir-126 is not responsible for the BLEC impairment, the following experiments analyse the signalling pathways egfl7 might be involved in.

Here the experimental data is correlative at best, in the case of Vegf, the conclusions are not at all supported by the data.

Major comments

The authors claim, that *egfl7* is essential for BLEC development as analysed by *lyve1b*:dsRed expression and provide in Figure 2 evidence, that there are no BLECs in *egfl7* mutants.

However in Figure 3, image c) there are multiple dsRed positive cells, which do not express GFP (after p2A mediated GFP overexpression to label *Egfl7* positive cells). Why?

Also Figure 3 Image c) 4dpf seems focused on a different area or level? with an unusual *lyve1b*:dsRed expression, 4dpf and 6dpf do not seem to be the same embryo. Please provide lower mag images in the supplement, together with the effects of *lyve1b* driven *egfl7* overexpression in wt embryos.

The figure urgently needs a comparable WT picture and an indication what a double lymphatic loop is supposed to be. The description of a lymphatic loop is misleading as the BLECs at this stage are only loosely connected cells, but do not form a lymphatic vessel with a lumen.

Figure 5 c, d is supposed to illustrate the lack of proliferation in BLECs of *egfl7* mutants, however, the chosen timepoints for analysis do not support this conclusion, as PCNA positive BLEC quantification already starts with an imbalance in number. The authors would have to show, that there are timepoints with the same number of BLECs specified and then would have to show, when (and how much) proliferation fails to occur. This might be better evaluated using BrdU.

Regarding the analysis of signaling pathways, the conclusions are not all supported by the data and even correlative data is weak.

Figure 6 is flawed in many ways. The *Vegfc* overexpression needs to be more stringently controlled and evaluated. e.g. There are no data presented, that *Vegfc* overexpression with a single heatshock does indeed stimulate proliferation of lymphatic endothelial cells (for that the number of PL cells in the trunk and the number of BLECs in the brain need to be evaluated with and without heatshock). Without a stimulating effect in the WT, the experiment is meaningless and no conclusions can be drawn from a "no-effect" result. In fact the data presented show no significant change in WT BLECs as analysed by loop structure with or without hs driven *VEGfC* expression, which clearly shows, that this is not the right assay. The conclusions are not all supported by the data, as the following contradictory statement indicates:

173 *Vegfc* overexpression resulted in the hyper-proliferation of venous-derived ECs
174 prominently in the HM (Fig. 6b). Even with the *egfl7* defect, the effect of *Vegfc* stimulation on the
175 venous hyper branch was not entirely blocked (Fig. 6c, d). Therefore, the *Vegfc* overexpression
176 experiment worked, and *Egfl7* did not mediate the *Vegfc*-driven proliferation in venous hyper-
177 sprouting in the trunk.

This is NOT shown in Fig 6b, or c,d respectively.

likewise the Interaction with a "novel" Integrin receptor is not really shown.

The manuscript urgently needs editing by a native English speaker, as it is full of grammatical errors and unfavourable word choices.

Minor comments

The transgenic line Tg(*egfl7*:YFP), and the *egfl7* mutant generated lack allele designations
Fig 3

box and figure legend should not declare "plasmid injections to generate mosaic transgenic line", but "mosaic transgenic embryos" (observe throughout the manuscript)

Reviewer #3

(Remarks to the Author)
See attached file.

Author Rebuttal letter:

Point by Point Response to the Reviewers

Response to the Reviewer #1

This study addresses the role of the endothelial-expressed secreted factor EGFL7 in development of BLECs (aka MuLECs, FGPs), lymphatic-related perivascular meningeal cells recently discovered in zebrafish. Although a variety of knockdown studies have suggested an important role for EGFL7 in endothelial cells, knockout mice and mutant zebrafish lack obvious vascular phenotypes, at least in part because of upregulation of compensatory factors by transcriptional adaptation. In this study, the authors re-confirm lack of blood or lymphatic vessel phenotypes in EGFL7 mutant zebrafish, but they also report that these mutants fail to form BLECs and go on to carry out additional studies aimed at understanding more about how EGFL7 promotes BLEC

development. Although the loss-of-BLEC phenotype is noteworthy, substantial problems with this manuscript reduce its impact and significance.

The evidence that lyve1b:dsred-positive BLECs are missing in the EGFL7 mutant appears solid, although the authors should have used additional methods to validate the actual absence of BLECs in these mutants besides loss of expression of a single transgene.

Response to this point: In the revised manuscript, to validate the actual absence of BLECs in the *egfl7* mutant, we have used another Tg(*prox1:kalTA4;UAS:TagRFP*) transgenic line labeling lymphatics to further demonstrate a significant reduction of the *prox1*+ BLECs in the *egfl7* mutant (Supplementary Fig. 3 a-d). In addition, the combinations of FISH with antibody staining have confirmed the decreased expression of *vegfr3* and *mrc1a* in the *egfl7* mutant (Fig. 2 f-j).

The data supporting many or most of the other conclusions in this manuscript is often much less solid. This includes extensive use of selective data such as individual magnified images shown with little or no anatomical context, subjective or missing quantification of phenotypes, lack of necessary controls, and/or data that do not clearly show what the authors are concluding in their text (eg, cell autonomy data in fig. 3, specification data in Fig. 4, proliferation data in Fig. 5).

Response to this point:

According to the reviewer's suggestion, we have applied zebrafish brain anatomical context to indicate the imaged areas for all the magnified images in the revised manuscript, and we have provided quantifications of phenotypes in the figures (Fig. 2 c-e; Fig. 3 e, j; Fig. 4 d, i, j; Fig. 5f; Fig. 6 f-h). Furthermore, the cell autonomy data in Fig. 3, specification data in Fig. 4, and proliferation data in Fig. 5 have been improved based on this reviewer and other reviewers's suggestions.

Many of the descriptions of the precise methods and reagents used that are provided in the text and legends are not adequate to fully understand exactly how the experiment was done.

Response to this point: We have provided detailed methods and reagents in the Methods of the revised manuscript. In the figures (Fig. 6a; Supplementary Fig. 5e; Supplementary Fig. 7a), we have included some schematic diagrams to show the experimental design.

Problems with grammar and usage throughout the manuscript sometimes also get in the way of a clear understanding of the descriptions provided.

Response to this point: We apologize for the language issues since none of the authors are English native speakers. In the revised manuscript, we have tried our best to edit the language and have the manuscript edited by a native English speaker.

The potential significance is also overstated - the authors purport to be studying "brain lymphangiogenesis" in their title and throughout the text but BLECs are neither in the brain (they are in the meninges surrounding the brain, not in the brain proper) nor are they lymphatic vessels (they are individual separated perivascular cells on the outside of meningeal blood vessels). The authors do not look at actual zebrafish meningeal lymphatics in this study. All of this substantially detracts from the impact and significance of this manuscript.

Response to this point: The reviewer is correct that the statement "brain lymphangiogenesis" is not precise. We have changed the title and statements accordingly.

Response to the Reviewer #2

The authors describe a novel role for *Egfl7* in regulating brain lymphatic endothelial cell (BLECs) development. They claim that *Egfl7* is essential for sprouting and proliferation of BLECs by signaling via Integrin α 3 and mediating ILK-Vegfr3 signaling.

First of all, the requirement for *Egfl7* in BLEC development is completely novel and therefore worthwhile of publication in Nature Communications. The authors show, that it is expressed in BLECs (as well as in blood vessel endothelial cells) and that loss of function impairs BLEC development, but not progenitor specification. Furthermore the newly generated *egfl7* mutants survive to adulthood without gross morphological consequences or lymphedema.

After arguing the *mir-126* is not responsible for the BLEC impairment, the following experiments analyse the signalling pathways *egfl7* might be involved in.

Here the experimental data is correlative at best, in the case of Vegf, the conclusions are not at all supported by the data.

Major comments

The authors claim, that *egfl7* is essential for BLEC development as analysed by *lyve1b:dsRed* expression and provide in Figure 2 evidence, that there are no BLECS in *egfl7* mutants. However in Figure 3, image c) there are multiple dsRed positive cells, which do not express GFP (after p2A mediated GFP overexpression to label *Egfl7* positive cells). Why? Also Figure 3 image c) 4 dpf seems focused on a different area or level? with an unusual *lyve1b:dsRed* expression, 4 dpf and 6 dpf do not seem to be the same embryo. Please provide lower mag images in the supplement, together with the effects of *lyve1b* driven *egfl7* overexpression in wt embryos.

Response to this point: In Figure 3 of the original manuscript, we used plasmid injection to generate mosaic Tg(*lyve1b:egfl7-p2A-GFP*) transgenic embryos for the replenishment of *Egfl7* in the LECs. Thus, the GFP expressed by LECs was mosaic, and not all the *lyve1b:DsRed* positive cells expressed GFP.

However, the mosaic overexpression of *Egfl7* in the BECs or LECs to rescue the lack of BLECs in the *egfl7* mutant might not be convincing enough. In the revised manuscript, we have generated stable transgenic lines. Using the Tg(*kdr1:GFP*) and Tg(*lyve1b:GFP*) lines as controls, replenishments of *Egfl7* either in BECs using the Tg(*kdr1:egfl7-p2A-GFP*) transgene or in LECs using the Tg(*lyve1b:egfl7-p2A-GFP*) transgene rescued the bilateral loop in *egfl7* mutants at 6 dpf (Fig. 3b, d, e, g, i, j). These results indicate that both the BEC-secreted and LEC-secreted *Egfl7* are functional to induce BLEC formation. In addition, as requested by the reviewer, we have also provided overexpression of *Egfl7* in BECs or LECs in the WT. Overexpression of *Egfl7* in BECs promotes more BLEC formation in the WT, while that in LECs exhibited weaker effects (Fig. 3a, c, f, h). This might be caused by more BECs than LECs in the meninges.

The figure urgently needs a comparable WT picture and an indication what a double lymphatic loop is supposed to be. The description of a lymphatic loop is misleading as the BLECs at this stage are only loosely connected cells, but do not form a lymphatic vessel with a lumen.

Response to this point: According to the reviewer's suggestion, in the new Figures 3a and 3f, we provided lower magnification images to show the bilateral loop-like structure in the WT brain. We agree that the description of a lymphatic loop is misleading as the BLECs at this stage are only loosely connected cells. We used the term lymphatic loop in the original manuscript only because it was used in a previous study (Bower., 2017). In the revised manuscript, to highlight the structure but avoid misleading, we have switched the term from lymphatic loop to loop-like structure.

Figure 5 c, d is supposed to illustrate the lack of proliferation in BLECS of *egfl7* mutants, however, the chosen timepoints for analysis do not support this conclusion, as PCNA positive BLEC quantification already starts with an imbalance in number. The authors would have to show, that there are timepoints with the same number of BLECs specified and then would have to show, when (and how much) proliferation fails to occur. This might be better evaluated using BrdU.

Response to this point: According to the reviewer's suggestion, we have performed EdU staining to label BLEC proliferation from 4 dpf to 6 dpf. We injected EdU into the brain and CCVs at 56 hpf, the time point when BLECs progenitors begin to specify. The EdU labelling indicated that the proliferating BLECs decreased from 4 dpf to 6 dpf in the mutant (Fig. 5 d-f and Supplementary Fig. 5 e-j).

Regarding the analysis of signaling pathways, the conclusions are not all supported by the data and even correlative data is weak.

Figure 6 is flawed in many ways. The *Vegfc* overexpression needs to be more stringently controlled and evaluated.

e.g. There are no data presented, that *Vegfc* overexpression with a single heatshock does indeed stimulate proliferation of lymphatic endothelial cells (for that the number of PL cells in the trunk and the number of BLECs in the brain need to be evaluated with and without heatshock). Without a stimulating effect in the WT, the experiment is meaningless and no conclusions can be drawn from a no-effect result. In fact the data presented show no significant change in WT BLECs as analysed by loop structure with or without hs driven VEGFc expression, which clearly shows, that this is not the right assay.

The conclusions are not all supported by the data, as the following contradictory statement indicates: 173 *Vegfc* overexpression resulted in the hyper-proliferation of venous-derived ECs 174 prominently in the HM (Fig. 6b). Even with the *egfl7* defect, the effect of *Vegfc* stimulation on the

175 venous hyper branch was not entirely blocked (Fig. 6c, d). Therefore, the Vegfc overexpression
176 experiment worked, and Egfl7 did not mediate the Vegfc-driven proliferation in venous hyper-
177 sprouting in the trunk.

This is NOT shown in Fig6b, or c,d respectively.

Response to this point: In this revised manuscript, we have included the injection of a plasmid containing hsp70l:Venus as a control. A plasmid containing hsp70l:Vegfc-P2A-Venus was injected to create an ectopic, mosaic expression of Vegfc in embryos. After heat-shock, the ectopically overexpression of Vegfc, but not the control Venus protein, resulted in the hyper-proliferation and hyper-branching of venous-derived ECs, as shown in the PLs and ISVs in both WT and mutant (Fig. 6a, b, d, f, g; Supplementary Fig. 7b, d, f). But in the brain, the ectopic expression of Vegfc could not rescue the lack of BLECs in the mutant, and the BLECs in the WT remained unaffected in contrast to the ectopic expression of Venus (Fig. 6c, e, h and Supplementary Fig. 7c, e, g). These observations suggest that Egfl7 does not act upstream of Vegfc in the regulation of BLEC formation. According to the reviewer's suggestion, we have deleted the inappropriate statement in line 174-177.

likewise the Interaction with a "novel" Integren receptor is not really shown.

Response to this point: According to the reviewer's suggestion, we have conducted coimmunoprecipitation experiments in cell culture to validate the physical association of EGFL7 with integrin $\alpha 5 \beta 3$ (Fig. 7i).

The manuscript urgently needs editing by a native English speaker, as it is full of grammatical errors and unfavourable word choices.

Response to this point: We apologize for the language issues. We have tried our best to edit the revised manuscript, and have it edited by a native English speaker.

Minor comments

The transgenic line Tg(egfl7:YFP), and the egfl7 mutant generated lack allele designations

Response to this point: According to the reviewer's suggestion, we have added the allele designations for the newly generated transgenic lines and mutants.

Fig3

box and figure legend should not declare âplasmid injections to generate mosaic transgenic lineâ, but âmosaic transgenic embryosâ (observe throughout the manuscript)

Response to this point: We have corrected the description in Figure 6. In the Figure 3 of the revised manuscript, stable transgenic lines have been used in the rescue experiments.

Response to the Reviewer #3

In the current manuscript, Ding et al describe Egfl7, a previously known angiogenic factor, to be required for the formation of BLECs. The authors generate a novel egfl7 zebrafish mutant characterized by the absence of the brain lymphatic loops, while other lymphatic vessels such as facial lymphatics and trunk lymphatics remain unaffected by this mutation. Using gain-of-function approaches as well as chemical approaches, they propose a potential role for Egfl7 in regulating BLECs formation through integrin $\alpha 5 \beta 3$ /ILK pathway. Overall, the study is interesting and explores a relevant topic in lymphatic development. However, after careful evaluation, I find that the work, while promising, falls short in several critical aspects that require further attention before it can be considered for publication.

Strengths: The manuscript's primary strength lies in its exploration of Brain lymphatic endothelial cells (BLECs), which represent a very unique population of lymphatic ECs that is poorly understood. The egfl7 mutant generated by the authors shows a complete reduction of BLECs while retaining trunk and facial lymphatics. If correct, this is an important finding as it identifies a very distinct mechanism for the formation of this highly specialized fate of lymphatic ECs. Furthermore, this mutant is of potential for studies on the functional evaluation of BLECs.

Major concerns:

Image Quality and Clarity: The figures and images provided in the manuscript do not meet the standard required for publication. Some of the images lack clarity, making it difficult for readers to comprehend the presented data. Additionally, insufficient labeling and incomplete explanations accompanying the images hinder the overall understanding of the study. The authors should enhance the image quality, provide better resolutions, and include detailed

captions to facilitate reader comprehension. In particular, in order to fully understand the phenotype, it is necessary to have comparable images of the same anatomical fields with the same magnification, resolution, and orientation.

Response to this point: According to the reviewer's suggestion, we have provided new images to replace the Figure panels in low quality or poor clarity, such as Figure 2a,k, and in Figures 3-7. We have also provided zebrafish brain anatomical context to indicate the imaged areas with comparable magnification, resolution, and orientation (Fig. 1e; Fig. 4a; Fig. 5a; Fig. 7a, b, g; Supplementary Fig. 1e; Supplementary Fig. 5a; Supplementary Fig. 6a; Supplementary Fig. 8a, d).

Limited Quantification: One of the major drawbacks of this manuscript is the lack of quantitative analysis and statistical validation of the phenotypes. While the qualitative insights presented are intriguing, the absence of rigorous quantitative measures hinders the ability to draw concrete conclusions and ascertain the significance of the results.

Response to this point: According to the reviewer's suggestion, we have provided quantifications for phenotypes in the figures (Fig. 2 c-e; Fig. 3 e, j; Fig. 4 d, i, j; Fig. 5f; Fig. 6 f-h).

1. The characterization of the mutant is quite poor. The characterization of such a mutant requires a deeper understanding of the defects in both the blood and lymphatic compartments. The main problem stems from the inconsistency of the images presented- some images give the impression that all the brain *lyve1+* cells are missing, while in others, clear *lyve1b+* vessels are observed. This is particularly relevant for Fig 4a,b and 6e-h. Further, it seems there is quite a degree of variability in the images shown. While in Fig.2, the wt vs mut blood vessels look comparable, in Fig 4c-d the blood vessels of mutants look different from the WT. If indeed there is variation in the blood vessels of the mutant, that should be reflected in the quantification provided before.

Response to this point: Excuse the confusions in the original manuscript. In the *egfl7* mutant, the BLEC progenitors could still be specified from the CVP, as shown by the emergence of *Prox1+/-vegfr3+/mrc1a+* BLECs around the CVP in the lower middle layer of the brain during the initial sprouting (as shown in Fig. 4, Fig. 5c, and Supplementary Movie. 1). However, these cells failed to migrate along the MsV to form the bilateral loop-like structure, and their proliferation was blocked, leading to the absence of a loop-like structure that covers the brain surface (as shown in the top layer of the brain in Fig. 2f-j). The total number of BLECs was reduced in the mutant as shown in Fig. 2a-j, Fig. 3b, g, and Fig. 5c, e.

Furthermore, there was no significant difference in the blood vessels in the brain between WT and the *egfl7* mutant. We would like to clarify that in the initial manuscript, the original Figure 4 c-d showed different layers of the brain and this may cause some confusions. In the revised manuscript, we have shown that the blood vessels are similar in the WT and mutant (Fig. 4e-h).

2. Time-lapse movies of loop formation in wt vs. *egfl7* mutants would be necessary in order to support the claims of defective sprouting/retraction. These points cannot be made based on snapshots of different embryos.

Response to this point: According to the reviewer's suggestion, we have provided the time-lapse movies from 66 hpf to 122 hpf to show the formation of the loop-like structure in WT vs *egfl7* mutant (Supplementary Movie. 1).

3. The claim that LEC differentiation is not affected is based on a single, extremely low-quality image of *prox1* staining shown in Fig.4. The orientation of this figure is very different from the rest. Have the authors detected 1 *prox1a+* EC in all their samples (n should be mentioned)? In 4c, it seems that the number of *Vegfr3+* ECs is much higher. Presumably, these should be *prox1+* also. Since *prox1* is also expressed in neighboring neurons, the authors should provide single slices from the scans to clearly indicate the colocalization between *prox1* nuclei and *lyve1*. Overall, the data provided are obviously not enough to sustain the proposed mechanism.

Response to this point: In the Fig. 4a of the revised manuscript, we have provided comparable schematic diagrams. The anti-*Prox1* staining were shown in lateral views of the brain (Fig. 4b, c), while FISH-*vegfr3* and FISH-*mrc1a* showed the lower middle layer of the brain in dorsal views (Fig. 4e-h). We chose the lateral view for assessing the expression of *Prox1* because it is also expressed in neurons, and the *Prox1+* neuron signals may interfere the BLECs signals in the deep layer of the brain. The ratios of *lyve1b+* BLECs also positive for *Prox1* is quantified for

both WT and *egfl7* mutant (Fig. 4d, n=9 embryos). We also provided 2D images in Fig. 4b-c, where the single slice showed that the *Prox1*⁺ nuclei merged with in the *lyve1b*⁺ BLECs. In Fig. 4e-h, the dorsal views of lower middle layer of the brain showed that the BLECs emerged around the CVPs. There are more *Vegfr3*⁺ BLECs visible in the dorsal view than in the lateral view.

4. An additional major problem is the fact that all this work is based on a single transgenic reporter- *lyve1b*. Hence, it is not possible to ascertain whether is the BLECs that are absent or the *lyve1* signal that is reduced. Since BLECs have been shown to be also labeled by *flt4* and *mrc1a*, it would be necessary to show that they are absent using more than one transgenic line.

Response to this point: In the revised manuscript, we have used another transgenic line *Tg(prox1:kaITTA4;UAS:TagRFP)* to label LECs and demonstrate the *prox1*⁺ BLECs were reduced in the *egfl7* mutant (Supplementary Fig. 3a-d). In addition, we have used combination of FISH and antibody staining to validate the reduced expression of *vegfr3* and *mrc1a* in the mutant (Fig. 2f-j).

5. Although the BLECs phenotype looks quite strong, based on the presented images it is clear that the mutant also displays defects in the facial lymphatics, specifically the OTL and LAA. The table presented in Fig. 2c does not provide a clear quantification of the phenotypes, but rather gives a simplistic binary (0% Or 100%) inference that all kinds of vessels are unaffected and there is a complete reduction of BLECs. However, it is clear from Fig 2a,b (6dpf, lateral head) that OLV, for instance is reduced in the mutants. The authors should provide clear quantifications describing the exact changes in the vasculature of *Egfl7*^{-/-}.

Response to this point: According to the reviewer's suggestion, we have provided clear quantifications of the facial lymphatics and TD in the WT and *egfl7* mutant (Fig. 2c-e and Supplementary Fig. 4e, h).

6. Given the controversy surrounding previous *egfl7* mutants and morphants it is absolutely necessary that the authors confirm the reduction in *egfl7* levels mRNA and protein.

Response to this point: In the revised manuscript, we have provided the data showing the reduction of *egfl7* mRNA in the *egfl7* mutant by real-time qPCR (Supplementary Fig. 2b). In addition, *Egfl7*^{mut}-Myc protein failed to express in the transfected H293T cells (Supplementary Fig. 2c).

7. The rescue experiments are essential to understand the putative role of *egfl7*. However, several points are unclear:

â€ First, technically, why use two different vectors (pBluescript vs pTol backbones) Are they equally efficient? For the images provided, it seems they might have expression differences. In any case, itâ€™s be advisable to repeat the experiments using stable lines to strengthen.

â€ Secondly, the mosaic injection of pBluescript-kdrl-*egfl7*-p2A-GFP show a major increase of *lyve1b*⁺ vessels in the *egfl7* mutant comparing it with the initial characterization. This would indicate that blood-derived *egfl7* could also have a role in BLEC formation.

â€ Finally, for the proposed model: it is unclear how BLECs can differentiate whether *egfl7* is produced by blood/lymphatic cells.

Response to this point:

Both the pBluescript and pTol vectors are efficient to generate transgenic lines. However, based on the reviewerâ€™s suggestion, we believed that carrying out the rescue experiments using transient mosaic overexpression of *Egfl7* is not convincing. Therefore, in the revised manuscript, we have used stable transgenic lines to perform the rescue experiments. We have also included *Tg(kdrl:GFP)* and *Tg(lyve1b:GFP)* transgenic lines as controls. In fact, we have obtained different conclusions in contrast to the original manuscript. The replenishment of *Egfl7* either in BECs using the *Tg(kdrl:egfl7-p2A-GFP)* transgene, or in LECs using the *Tg(lyve1b:egfl7-p2A-GFP)* transgene, could rescue the bilateral loop-like structures in the *egfl7* mutants (Fig. 3b, d, e, g, i, j). Moreover, we have performed overexpression of *Egfl7* in BECs or LECs in the WT. Overexpression of *Egfl7* in BECs promoted more ectopic BLEC formation than overexpression of *Egfl7* in LECs (Fig. 3a, c, f, h).

This is because of more meningeal blood vessels than BLECs in the top layer, so that overexpression in BECs could produce more *Egfl7* protein. These results indicate that both the vascular-derived and lymphatic-derived *Egfl7* play roles in BLEC formation.

For the proposed model (Supplementary Fig. 9), we have corrected the *Egfl7* origin during BLEC formation.

8. The ILK â€ *Vegfr3* link requires further direct evidence to substantiate the proposed model.

Have the authors checked if ILK inhibition through cpd22 inhibitor, phenocopies some of the Vegfc overexpression data (such as those in 6b,c)?

According to the reviewer's suggestion, we have applied the cpd22-ilk inhibitor, which was ineffective to the trunk lymphangiogenesis and could not phenocopy the Vegfc overexpression data (Fig. 6d). This could be explained by the dependence of trunk lymphangiogenesis primarily on the Vegfc-Vegfr3 signaling, rather than Egfl7-integrin signaling. Because trunk lymphangiogenesis remains normal in the egfl7 mutant (Fig. 2b), which means that the Egfl7-integrin signaling is not required for trunk lymphangiogenesis.

In the revised manuscript, we have provided a new proposed model to illustrate the mechanisms underpinning the regulation of BLEC formation by Egfl7 (Supplementary Fig. 9). In the WT, Egfl7 is deposited in the ECM upon secretion from BECs and LECs. Then, Egfl7 activates Integrin $\alpha 5 \beta 3$, which disrupts the Integrin-ILK association and increases the Vegfr3 phosphorylation, in turn promotes BLEC proliferation. In the egfl7 mutant, ILK interacts with Integrin $\alpha 5 \beta 3$, and Vegfr3-Integrin $\alpha 5 \beta 3$ dissociates, thus reducing Vegfr3 phosphorylation and inhibiting BLEC proliferation.

Minor concerns

• N number should be increased to increase statistical power.

• Some of the graphs do not correlate with the message and the representative images.

• It is surprising that in Supplementary Fig.4, the authors do not find any apoptotic cells in the region of the brain depicted. Have the authors verified that the TUNEL assay is working well in these samples?

Response to this point: According to the reviewer's suggestion, we have increased the n numbers in the statistics. We have improved the entire article content to ensure the correlation of graphs with the message and the representative images. We have provided positive controls of TUNEL assays. Supplementary Fig. 6 showed that BLEC apoptosis was hardly detectable in the WT and egfl7 mutants.

Overall, this work appears to be at a preliminary stage. While it has the potential to contribute to the field, the manuscript requires significant work to address the mentioned limitations.

Response to this point: Hopefully we have addressed all the mentioned limitations above in the revised manuscript. Thanks for the helpful comments to improve the quality of our work.

Version 1:

Reviewer comments:

Reviewer #4

(Remarks to the Author)

The revised manuscript by Chen et al. has addressed the major concerns raised by the three reviewers. Through the addition of clearer and more comprehensive data, the authors have provided strong support for their main conclusion that Egfl7 is specifically required for the development of BLECs. Overall, the manuscript has significantly improved, and it is suitable for publication in Nature Communications.

Reviewer #5

(Remarks to the Author)

The authors have addressed most concerns raised by the reviewers. Additional minor concerns need to be addressed before this manuscript can be accepted. Specifically, the authors should have the manuscript edited by a native English speaker to avoid confusions.

Specific comments:

The authors provided new data for egfl7 mutant characterization in Figure Supplemental 2b and 2c. These data are supposed to show that mutation in egfl7 caused the non-sense mediated decay and absence of protein expression. In Supplemental Figure 2b, the egfl7 transcripts seem to be only slightly degraded. And the western blot in supplemental Figure 2C using HEK293 cell is confusing. The authors can elaborate more on how they did the experiments in methods or legends. If a truncated Egfl7 protein is present, the western blot gel might not be able to detect it since it is much smaller. Furthermore, it is still not possible to rule out that the expression construct of Egfl7 mutant simply did not work. These are some caveats but the authors can clarify it to help interpret the results or difference between mutants and morphants.

Minor comment: Line 61—EMI not defined

Reviewer #6

(Remarks to the Author)

The paper by Chen et al provides convincing evidence that Egfl7 regulates the formation of the BLECs of the zebrafish brain. The phenotypic studies are convincing and the authors have generally done a good job of responding to the reviewers concerns. The study will be of interest to vascular biologists, those interested in lymphatic development, lymphangiogenesis and also to neuroscientists working on mural cell populations in the brain.

While the authors have mostly responded will to the concerns of the reviewers, I have some concerns over the final figure of the manuscript.

In FIGURE 7 it is proposed that downstream changes in integrin signalling involving Itgb3b and Itgav as well as Ilk are responsible for the Egfl7 phenotype. The mechanistic data here are somewhat cursory and require additional lines of evidence. Much rests on the fact that the inhibitor Cilengitide causes a loss of BLECs, but this is not backed up with any additional methods such as the analysis of a genetic model of loss of Itgb3b or Itgav. It would be more rigorous and appropriate to back this up with at least one additional model (a complementary drug treatment or mutant).

Furthermore, the inhibitor Cpd22 is used to show that loss of Ilk can partially rescue the Egfl7 phenotype. This is more convincing because it rescues a genetic mutant, but again it would be appropriate and more rigorous to support this with an additional line of evidence. Using the Ilk mutant (that is published) to see if Ilk genetically interacts with Egfl7 in BLEC formation would be a good experiment here.

Strengthening this mechanism is in my view needed given the strength of the conclusions being drawn by the authors.

Author Rebuttal letter:

Point by Point Response to the Reviewers

Response to the Reviewer #4

Reviewer #4 (Remarks to the Author):

The revised manuscript by Chen et al. has addressed the major concerns raised by the three reviewers. Through the addition of clearer and more comprehensive data, the authors have provided strong support for their main conclusion that Egfl7 is specifically required for the development of BLECs. Overall, the manuscript has significantly improved, and it is suitable for publication in Nature Communications.

Response to this point: Thanks for the reviewer's affirmation.

Response to the Reviewer #5

Reviewer #5 (Remarks to the Author):

The authors have addressed most concerns raised by the reviewers. Additional minor concerns need to be addressed before this manuscript can be accepted. Specifically, the authors should have the manuscript edited by a native English speaker to avoid confusions.

Specific comments:

The authors provided new data for egfl7 mutant characterization in Figure Supplemental 2b and 2c. These data are supposed to show that mutation in egfl7 caused the non-sense mediated decay and absence of protein expression. In Supplemental Figure 2b, the egfl7 transcripts seem to be only slightly degraded. And the western blot in supplemental Figure 2C using HEK293 cell is confusing. The authors can elaborate more on how they did the experiments in methods or legends. If a truncated Egfl7 protein is present, the western blot gel might not be able to detect it since it is much smaller. Furthermore, it is still not possible to rule out that the expression construct of Egfl7 mutant simply did not work. These are some caveats but the authors can clarify it to help interpret the results or difference between mutants and morphants.

Response to this point: In the revised manuscript, we have provided a detailed description of the methodology and legend (Supplementary Figure 2) used for creating the constructs and conducting the experiments. Essentially, we used cDNAs encoding Egfl7WT (831 bp/277 aa) and Egfl7cq180 (213 bp/71 aa) proteins that were PCR-amplified using cDNA from WT or egfl7^{-/-} embryos. Following cloning and cell transfection, we detected the expression of the C-terminally tagged Egfl7WT-Myc protein (about 31 kDa) and Egfl7cq180-Myc protein (about 8 kDa) in HEK293T cells. Western Blot analysis revealed that the Egfl7-WT-Myc protein was expressed highly in the cell lysates. However, the truncated Egfl7-mutant-myc protein was strongly decreased; if we increased the exposure, the slight expression of Egfl7-mutant-myc protein could be seen in the western blot gel (Response Figure 1, arrow). It is worth mentioning that Egfl7cq180 shares high similarity to the truncated protein that was produced in the Egfl7s981 mutant embryos. In

those embryos, the protein was either strongly reduced or undetectable in the cells (Rossi et al., 2015).

Response Figure 1. The slight expression of Egfl7-mutant-myc protein could be detected in the cell lysates (arrow) after increased the exposure of Supplementary Fig. 2c.

Minor comment: Line 61 EMI not defined

Response to this point: According to the reviewer's suggestion, we have added the definition of the EMI.

Response to the Reviewer #6

Reviewer #6 (Remarks to the Author):

The paper by Chen et al provides convincing evidence that Egfl7 regulates the formation of the BLECs of the zebrafish brain. The phenotypic studies are convincing and the authors have generally done a good job of responding to the reviewers concerns. The study will be of interest to vascular biologists, those interested in lymphatic development, lymphangiogenesis and also to neuroscientists working on mural cell populations in the brain.

While the authors have mostly responded will to the concerns of the reviewers, I have some concerns over the final figure of the manuscript.

In FIGURE 7 it is proposed that downstream changes in integrin signalling involving Itgb3b and Itgav as well as Ilk are responsible for the Egfl7 phenotype. The mechanistic data here are somewhat cursory and require additional lines of evidence. Much rests on the fact that the inhibitor Cilengitide causes a loss of BLECs, but this is not backed up with any additional methods such as the analysis of a genetic model of loss of Itgb3b or Itgav. It would be more rigorous and appropriate to back this up with at least one additional model (a complementary drug treatment or mutant).

Response to this point: According to the reviewer's suggestion, we applied CRISPRi technology (dCas9-KRAB) to inhibit itgb3b transcript elongation. The co-injection of dCas9 mRNA and itgb3b gRNAs could reduce the number of BLECs in the brain (Figure 7m-p). This additional knockdown approach further demonstrates Egfl7 interacts with integrin $\beta 3$ to regulate the BLEC development.

Furthermore, the inhibitor Cpd22 is used to show that loss of Ilk can partially rescue the Egfl7 phenotype. This is more convincing because it rescues a genetic mutant, but again it would be appropriate and more rigorous to support this with an additional line of evidence. Using the Ilk mutant (that is published) to see if Ilk genetically interacts with Egfl7 in BLEC formation would be a good experiment here.

Response to this point: According to the reviewer's suggestion, we have generated the ilk mutant (Supplementary Figure 9) and used the ilk heterozygous to conduct the rescue experiment as the ilk mutant exhibited severe deformities. The findings revealed that the egfl7 mutant's absence of loop-like structures was partially rescued when crossed with the ilk heterozygous mutant (Figure 8h-k).

Strengthening this mechanism is in my view needed given the strength of the conclusions being drawn by the authors.

Response to this point: Thank you for your helpful comments on ways to improve the quality of our work.

Version 2:

Reviewer comments:

Reviewer #6

(Remarks to the Author)

The authors have responded to all of my concerns relating to the proposed molecular mechanism. They show with new data that itgb3b knockdown leads to loss of BLECs and that egfl7 mutants genetically interact with ilk heterozygotes. This new data strengthens the pharmacological data already provided and gives further confidence in the proposed mechanism. The manuscript convincingly demonstrates a role for egfl7 in BLEC development and suggests an interesting underlying mechanism.

I have no further concerns.

Author Rebuttal letter:

Point by Point Response to the Reviewers

Response to the Reviewer #6

Reviewer #6 (Remarks to the Author):

The authors have responded to all of my concerns relating to the proposed molecular mechanism. They show with new data that *itgb3b* knockdown leads to loss of BLECs and that *egfl7* mutants genetically interact with *ilk* heterozygotes. This new data strengthens the pharmacological data already provided and gives further confidence in the proposed mechanism. The manuscript convincingly demonstrates a role for *egfl7* in BLEC development and suggests an interesting underlying mechanism.

I have no further concerns.

Response to this point: Thanks for the reviewer's affirmation. We also thank the referees for the comments; it is very helpful and has considerably improved the quality of our work.
